# Solving Inverse Problems with Latent Diffusion Models via Hard Data Consistency

**Bowen Song**[1][\*], **Soo Min Kwon**[1][\*], **Zecheng Zhang**[2], **Xinyu Hu**[3], **Qing Qu**[1] **Liyue Shen**[1]

[1]University of Michigan, [2]Kumo.AI, [3]Microsoft

## Abstract

Latent diffusion models have been demonstrated to generate high-quality images, while offering efficiency in model training compared to diffusion models operating in the pixel space. However, incorporating latent diffusion models to solve inverse problems remains a challenging problem due to the nonlinearity of the encoder and decoder. To address these issues, we propose *ReSample*, an algorithm that can solve general inverse problems with pre-trained latent diffusion models. Our algorithm incorporates data consistency by solving an optimization problem during the reverse sampling process, a concept that we term as hard data consistency. Upon solving this optimization problem, we propose a novel resampling scheme to map the measurement-consistent sample back onto the noisy data manifold and theoretically demonstrate its benefits. Lastly, we apply our algorithm to solve a wide range of linear and nonlinear inverse problems in both natural and medical images, demonstrating that our approach outperforms existing state-of-the-art approaches, including those based on pixel-space diffusion models.

## 1 Introduction

Inverse problems arise from a wide range of applications across many domains, including computational imaging (Beck & Teboulle, 2009; Afonso et al., 2011), medical imaging (Suetens, 2017; Ravishankar et al., 2019), and remote sensing (Liu et al., 2021; 2022), to name a few. When solving these inverse problems, the goal is to reconstruct an unknown signal $x_* \in \mathbb{R}^n$ given observed measurements $y \in \mathbb{R}^m$ of the form

$$y = \mathcal{A}(x_*) + \eta,$$

where $\mathcal{A}(\cdot) : \mathbb{R}^n \to \mathbb{R}^m$ denotes some forward measurement operator (can be linear or nonlinear) and $\eta \in \mathbb{R}^m$ is additive noise. Usually, we are interested in the case when $m < n$, which follows many real-world scenarios. When $m < n$, the problem is ill-posed and some kind of regularizer (or prior) is necessary to obtain a meaningful solution.

In the literature, the traditional approach of using hand-crafted priors (e.g. sparsity) is slowly being replaced by rich, learned priors such as deep generative models. Recently, there has been a lot of interests in using diffusion models as structural priors due to their state-of-the-art performance in image generation (Dhariwal & Nichol, 2021a; Karras et al., 2022; Song et al., 2023b; Lou & Ermon, 2023a). Compared to generative adversarial networks (GANs), diffusion models are generally easier and more stable to train, making them a generative prior that is more readily accessible (Dhariwal & Nichol, 2021b). The most common approach for using diffusion models as priors is to resort to posterior sampling, which has been extensively explored in the literature (Song et al., 2022; Chung et al., 2023a; 2022; Kawar et al., 2022; Song et al., 2023a; Chung et al., 2023b; Meng & Kabashima, 2022; Zhang & Zhou, 2023). However, despite their remarkable success, these techniques exhibit several limitations. The primary challenge is that the majority of existing works train these models directly in the pixel space, which requires substantial computational resources and a large volume of training data (Rombach et al., 2022).

Latent diffusion models (LDMs), which embed data in order to operate in a lower-dimensional space, present a potential solution to this challenge, along with considerable improvements in computational efficiency (Rombach et al., 2022; Vahdat et al., 2021) by training diffusion models in a

---

[\*]Equal Contribution; Corresponding authors {bowenbw, kwonsm}@umich.edu

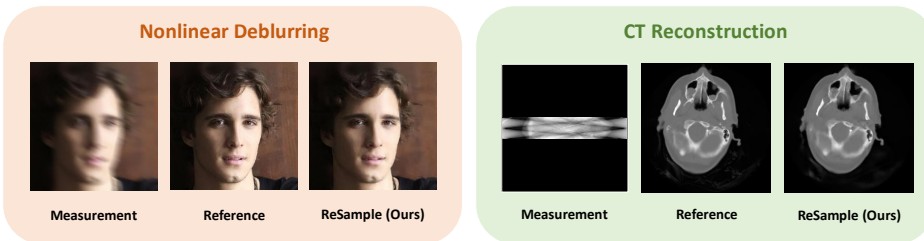

Figure 1: **Example reconstructions of our algorithm (*ReSample*) on two noisy inverse problems, nonlinear deblurring and CT reconstruction, on natural and medical images, respectively.**

compressed latent space. They can also provide a great amount of flexibility, as they can enable one to transfer and generalize these models to different domains by fine-tuning on small amounts of training data (Ruiz et al., 2023). Nevertheless, using LDMs to solve inverse problems poses a significant challenge. The main difficulty arises from the inherent nonlinearity and nonconvexity of the decoder, making it challenging to directly apply existing solvers designed for pixel space. To address this issue, a concurrent work by Rout et al. (2023) recently introduced a posterior sampling algorithm operating in the latent space (PSLD), designed to solve *linear* inverse problems with provable guarantees. However, we observe that PSLD may reconstruct images with artifacts in the presence of measurement noise. Therefore, developing an efficient algorithm capable of addressing these challenges remains an open research question.

In this work, we introduce a novel algorithm named *ReSample*, which effectively employs LDMs as priors for solving *general* inverse problems. Our algorithm can be viewed as a two-stage process that incorporates data consistency by (1) solving a hard-constrained optimization problem, ensuring we obtain the correct latent variable that is consistent with the observed measurements, and (2) employing a carefully designed resampling scheme to map the measurement-consistent sample back onto the correct noisy data manifold. As a result, we show that our algorithm can achieve state-of-the-art performance on various inverse problem tasks and different datasets, compared to existing algorithms. Notably, owing to using the latent diffusion models as generative priors, our algorithm achieves a reduction in memory complexity. Below, we highlight some of our key contributions.

- We propose a novel algorithm that enables us to leverage latent diffusion models for solving general inverse problems (linear and nonlinear) through hard data consistency.

- Particularly, we carefully design a stochastic resampling scheme that can reliably map the measurement-consistent samples back onto the noisy data manifold to continue the reverse sampling process. We provide a theoretical analysis to further demonstrate the superiority of the proposed stochastic resampling technique.

- With extensive experiments on multiple tasks and various datasets, encompassing both natural and medical images, our proposed algorithm achieves state-of-the-art performance on a variety of linear and nonlinear inverse problems.

## 2 BACKGROUND

**Denoising Diffusion Probabilistic Models.** We first briefly review the basic fundamentals of diffusion models, namely the denoising diffusion probabilistic model (DDPM) formulation (Ho et al., 2020). Let $\boldsymbol{x}_0 \sim p_{\text{data}}(\boldsymbol{x})$ denote samples from the data distribution. Diffusion models start by progressively perturbing data to noise via Gaussian kernels, which can be written as the variance-preserving stochastic differential equation (VP-SDE) (Song et al., 2021) of the form

$$d\boldsymbol{x} = -\frac{\beta_t}{2}\boldsymbol{x}dt + \sqrt{\beta_t}d\boldsymbol{w}, \tag{1}$$

where $\beta_t \in (0, 1)$ is the noise schedule that is a monotonically increasing sequence of $t$ and $\boldsymbol{w}$ is the standard Wiener process. This is generally defined such that we obtain the data distribution when $t = 0$ and obtain a Gaussian distribution when $t = T$, i.e. $\boldsymbol{x}_T \sim \mathcal{N}(\boldsymbol{0}, \boldsymbol{I})$. The objective of diffusion

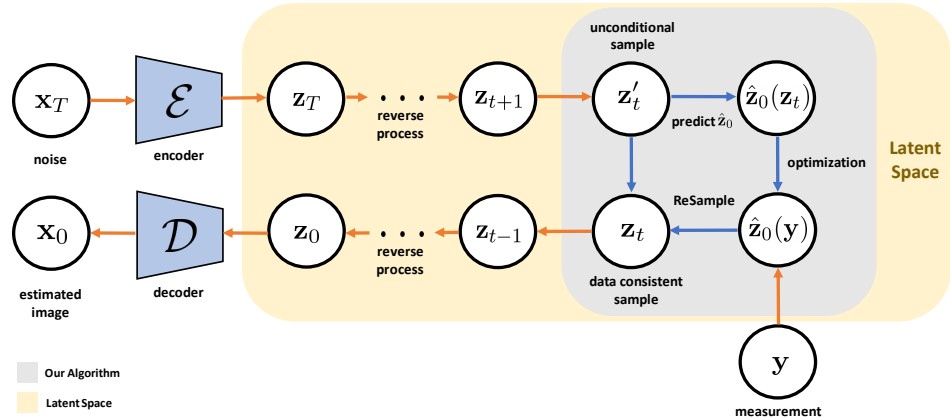

Figure 2: **Overview of our *ReSample* algorithm during the reverse sampling process conditioned on the data constraints from measurement.** The entire sampling process is conducted in the latent space upon passing the sample through the encoder. The proposed algorithm performs hard data consistency at some time steps $t$ via a skipped-step mechanism.

models is to learn the corresponding reverse SDE of Equation (1), which is of the form

$$dx = \left[ -\frac{\beta_t}{2} x - \beta_t \nabla_{x_t} \log p(x_t) \right] dt + \sqrt{\beta_t} d\bar{w}, \qquad (2)$$

where $d\bar{w}$ is the standard Wiener process running backward in time and $\nabla_{x_t} \log p(x_t)$ is the (Stein) score function. In practice, we approximate the score function using a neural network $s_\theta$ parameterized by $\theta$, which can be trained via denoising score matching (Vincent, 2011):

$$\hat{\theta} = \arg\min_{\theta} \mathbb{E} \left[ \| s_\theta(x_t, t) - \nabla_{x_t} \log p(x_t | x_0) \|_2^2 \right], \qquad (3)$$

where $t$ is uniformly sampled from $[0, T]$ and the expectation is taken over $t$, $x_t \sim p(x_t | x_0)$, and $x_0 \sim p_{\text{data}}(x)$. Once we have access to the parameterized score function $s_\theta$, we can use it to approximate the reverse-time SDE and simulate it using numerical solvers (e.g. Euler-Maruyama).

**Denoising Diffusion Implicit Models.** As the DDPM formulation is known to have a slow sampling process, Song et al. (2020) proposed denoising diffusion implicit models (DDIMs) that defines the diffusion process as a non-Markovian process to remedy this (Ho et al., 2020; Song et al., 2020; 2023b; Lu et al., 2022). This enables a faster sampling process with the sampling steps given by

$$x_{t-1} = \sqrt{\bar{\alpha}_{t-1}} \hat{x}_0(x_t) + \sqrt{1 - \bar{\alpha}_{t-1} - \eta \delta_t^2} s_\theta(x_t, t) + \eta \delta_t \epsilon, \quad t = T, \dots, 0, \qquad (4)$$

where $\alpha_t = 1 - \beta_t$, $\bar{\alpha}_t = \prod_{i=1}^t \alpha_i$, $\epsilon \sim \mathcal{N}(0, I)$, $\eta$ is the temperature parameter, $\delta_t$ controls the stochasticity of the update step, and $\hat{x}_0(x_t)$ denotes the predicted $x_0$ from $x_t$ which takes the form

$$\hat{x}_0(x_t) = \frac{1}{\sqrt{\bar{\alpha}_t}} (x_t + \sqrt{1 - \bar{\alpha}_t} s_\theta(x_t, t)), \qquad (5)$$

which is an application of Tweedie's formula. Here, $s_\theta$ is usually trained using the epsilon-matching score objective (Song et al., 2020). We use DDIM as the backbone of our algorithm and show how we can leverage these update steps for solving inverse problems.

**Solving Inverse Problems with Diffusion Models.** Given measurements $y \in \mathbb{R}^m$ from some forward measurement operator $\mathcal{A}(\cdot)$, we can use diffusion models to solve inverse problems by replacing the score function in Equation (2) with the conditional score function $\nabla_{x_t} \log p(x_t | y)$. Then by Bayes rule, notice that we can write the conditional score as

$$\nabla_{x_t} \log p(x_t | y) = \nabla_{x_t} \log p(x_t) + \nabla_{x_t} \log p(y | x_t).$$

This results in the reverse SDE of the form

$$dx = \left[ -\frac{\beta_t}{2} x - \beta_t (\nabla_{x_t} \log p(x_t) + \nabla_{x_t} \log p(y | x_t)) \right] dt + \sqrt{\beta_t} d\bar{w}.$$

In the literature, solving this reverse SDE is referred as *posterior sampling*. However, the issue with posterior sampling is that there does not exist an analytical formulation for the likelihood term $\nabla_{\boldsymbol{x}_t} \log p(\boldsymbol{y}|\boldsymbol{x}_t)$. To resolve this, there exists two lines of work: (1) to resort to alternating projections onto the measurement subspace to avoid using the likelihood directly (Chung et al., 2022; Kawar et al., 2022; Wang et al., 2022) and (2) to estimate the likelihood under some mild assumptions (Chung et al., 2023a; Song et al., 2023a). For example, Chung et al. (2023a) proposed diffusion posterior sampling (DPS) that uses a Laplacian approximation of the likelihood, which results in the discrete update steps

$$\boldsymbol{x}_{t-1} = \sqrt{\bar{\alpha}_{t-1}}\hat{\boldsymbol{x}}_0(\boldsymbol{x}_t) + \sqrt{1 - \bar{\alpha}_{t-1} - \eta\delta_t^2}\boldsymbol{s}_\theta(\boldsymbol{x}_t, t) + \eta\delta_t\boldsymbol{\epsilon} \tag{6}$$

$$\boldsymbol{x}_{t-1} = \boldsymbol{x}'_{t-1} - \zeta\nabla_{\boldsymbol{x}_t}\|\boldsymbol{y} - \mathcal{A}(\hat{\boldsymbol{x}}_0(\boldsymbol{x}_t))\|_2^2, \tag{7}$$

where $\zeta \in \mathbb{R}$ can be viewed as a tunable step-size. However, as previously mentioned, these techniques have limited applicability for real-world problems as they are all built on the pixel space.

**Solving Inverse Problems with Latent Diffusion Models.** The limited applicability of pixel-based diffusion models can be tackled by alternatively utilizing more efficient LDMs as generative priors. The setup for LDMs is the following: given an image $\boldsymbol{x} \in \mathbb{R}^n$, we have an encoder $\mathcal{E} : \mathbb{R}^n \to \mathbb{R}^k$ and a decoder $\mathcal{D} : \mathbb{R}^k \to \mathbb{R}^n$ where $k \ll n$. Let $\boldsymbol{z} = \mathcal{E}(\boldsymbol{x}) \in \mathbb{R}^k$ denote the embedded samples in the latent space. One way of incorporating LDMs to solve inverse problems would be to replace the update steps in Equation (6) with

$$\boldsymbol{z}'_{t-1} = \sqrt{\bar{\alpha}_{t-1}}\hat{\boldsymbol{z}}_0(\boldsymbol{z}_t) + \sqrt{1 - \bar{\alpha}_{t-1} - \eta\delta_t^2}\boldsymbol{s}_\theta(\boldsymbol{z}_t, t) + \eta\delta_t\boldsymbol{\epsilon}, \tag{8}$$

$$\boldsymbol{z}_{t-1} = \boldsymbol{z}'_{t-1} - \zeta\nabla_{\boldsymbol{z}_t}\|\boldsymbol{y} - \mathcal{A}(\mathcal{D}(\hat{\boldsymbol{z}}_0(\boldsymbol{z}_t)))\|_2^2, \tag{9}$$

After incorporating LDMs, this can be viewed as a non-linear inverse problem due to the non-linearity of the decoder $\mathcal{D}(\cdot)$. As this builds upon the idea behind DPS, we refer to this algorithm as *Latent-DPS*. While this formulation seems to work, we empirically observe that Latent-DPS often produces reconstructions that are often noisy or blurry and inconsistent with the measurements. We conjecture that since the forward operator involving the decoder is highly nonconvex, the gradient update may lead towards a local minimum. We provide more insights in Appendix Section D.

## 3 RESAMPLE: INVERSE PROBLEMS USING LATENT DIFFUSION MODELS

### 3.1 PROPOSED METHOD

**Hard Data Consistency.** Similar to Latent-DPS, our algorithm involves incorporating data consistency into the reverse sampling process of LDMs. However, rather than a gradient update as shown in Equation (9), we propose to solve an optimization problem on *some time steps* $t$:

$$\hat{\boldsymbol{z}}_0(\boldsymbol{y}) \in \arg\min_{\boldsymbol{z}} \frac{1}{2}\|\boldsymbol{y} - \mathcal{A}(\mathcal{D}(\boldsymbol{z}))\|_2^2, \tag{10}$$

where we denote $\hat{\boldsymbol{z}}_0(\boldsymbol{y})$ as the sample consistent with the measurements $\boldsymbol{y} \in \mathbb{R}^m$. This optimization problem has been previously explored in other works that use GANs for solving inverse problems, and can be efficiently solved using iterative solvers such as gradient descent (Bora et al., 2017; Jalal et al., 2021; Shah et al., 2021; Lempitsky et al., 2018). However, it is well known that solving this problem starting from a random initial point may lead to unfavorable local minima (Bora et al., 2017). To address this, we solve Equation (10 starting from an initial point $\hat{\boldsymbol{z}}_0(\boldsymbol{z}_{t+1})$, where $\hat{\boldsymbol{z}}_0(\boldsymbol{z}_{t+1})$ is the estimate of ground-truth latent vector at time 0 based on the sample at time $t + 1$. The intuition behind this initialization is that we want to start the optimization process within local proximity of the global solution of Equation (10), to prevent resulting in a local minimum. We term this overall concept as *hard data consistency*, as we strictly enforce the measurements via optimization, rather than a "soft" approach through gradient update like Latent-DPS. To obtain $\hat{\boldsymbol{z}}_0(\boldsymbol{z}_{t+1})$, we use Tweedie's formula (Efron, 2011) that gives us an approximation of the posterior mean which takes the following formula:

$$\hat{\boldsymbol{z}}_0(\boldsymbol{z}_t) = \mathbb{E}[\boldsymbol{z}_0|\boldsymbol{z}_t] = \frac{1}{\sqrt{\bar{\alpha}_t}}(\boldsymbol{z}_t + (1 - \bar{\alpha}_t)\nabla \log p(\boldsymbol{z}_t)). \tag{11}$$

---

**Algorithm 1** ReSample: Solving Inverse Problems with Latent Diffusion Models

---

**Require:** Measurements $\boldsymbol{y}$, $\mathcal{A}(\cdot)$, Encoder $\mathcal{E}(\cdot)$, Decoder $\mathcal{D}(\cdot)$, Score function $\boldsymbol{s}_\theta(\cdot, t)$, Pretrained
LDM Parameters $\beta_t$, $\bar{\alpha}_t$, $\eta$, $\delta$, Hyperparameter $\gamma$ to control $\sigma_t^2$, Time steps to perform resample $C$
 $\boldsymbol{z}_T \sim \mathcal{N}(\boldsymbol{0}, \boldsymbol{I})$            ▷ Initial noise vector
 **for** $t = T - 1, \ldots, 0$ **do**
  $\boldsymbol{\epsilon}_1 \sim \mathcal{N}(\boldsymbol{0}, \boldsymbol{I})$
  $\hat{\boldsymbol{\epsilon}}_{t+1} = \boldsymbol{s}_\theta(\boldsymbol{z}_{t+1}, t+1)$          ▷ Compute the score
  $\hat{\boldsymbol{z}}_0(\boldsymbol{z}_{t+1}) = \frac{1}{\sqrt{\bar{\alpha}_{t+1}}}(\boldsymbol{z}_{t+1} - \sqrt{1 - \bar{\alpha}_{t+1}}\hat{\boldsymbol{\epsilon}}_{t+1})$    ▷ Predict $\hat{\boldsymbol{z}}_0$ using Tweedie's formula
  $\boldsymbol{z}'_t = \sqrt{\bar{\alpha}_t}\hat{\boldsymbol{z}}_0(\boldsymbol{z}_{t+1}) + \sqrt{1 - \bar{\alpha}_t - \eta\delta^2}\hat{\boldsymbol{\epsilon}}_{t+1} + \eta\delta\boldsymbol{\epsilon}_1$    ▷ Unconditional DDIM step
  **if** $t \in C$ **then**              ▷ ReSample time step
   $\hat{\boldsymbol{z}}_0(\boldsymbol{y}) \in \arg\min_{\boldsymbol{z}} \frac{1}{2}\|\boldsymbol{y} - \mathcal{A}(\mathcal{D}(\boldsymbol{z}))\|_2^2$    ▷ Solve with initial point $\hat{\boldsymbol{z}}_0(\boldsymbol{z}_{t+1})$
   $\boldsymbol{z}_t = \text{StochasticResample}(\hat{\boldsymbol{z}}_0(\boldsymbol{y}), \boldsymbol{z}'_t, \gamma)$       ▷ Map back to $t$
  **else**
   $\boldsymbol{z}_t = \boldsymbol{z}'_t$         ▷ Unconditional sampling if not resampling
 $\boldsymbol{x}_0 = \mathcal{D}(\boldsymbol{z}_0)$          ▷ Output reconstructed image

---

However, we would like to note that performing hard data consistency on every reverse sampling iteration $t$ may be very costly. To address this, we first observe that as we approach $t = T$, the estimated $\hat{\boldsymbol{z}}_0(\boldsymbol{z}_{t+1})$ can deviate significantly from the ground truth. In this regime, we find that hard data consistency provides only marginal benefits. Additionally, in the literature, existing works point out the existence of a three-stage phenomenon (Yu et al., 2023), where they demonstrate that data consistency is primarily beneficial for the semantic and refinement stages (the latter two stages when $t$ is closer to 0) of the sampling process. Following this reasoning, we divide $T$ into three sub-intervals and only apply the optimization in the latter two intervals. This approach provides both computational efficiency and accurate estimates of $\hat{\boldsymbol{z}}_0(\boldsymbol{z}_{t+1})$.

Furthermore, even during these two intervals, we observe that we do not need to solve the optimization problem on every iteration $t$. Because of the continuity of the sampling process, after each data-consistency optimization step, the samples in the following steps can retain similar semantic or structural information to some extent. Thus, we "reinforce" the data consistency constraint during the sampling process via a skipped-step mechanism. Empirically, we see that it is sufficient to perform this on every 10 (or so) iterations of $t$. One can think of hard data consistency as guiding the sampling process towards the ground truth signal $\boldsymbol{x}^*$ (or respectively $\boldsymbol{z}^*$) such that it is consistent with the given measurements. Lastly, in the presence of measurement noise, minimizing Equation (10) to zero loss can lead to overfitting the noise. To remedy this, we perform early stopping, where we only minimize up to a threshold $\tau$ based on the noise level. We will discuss the details of the optimization process in the Appendix. We also observe that an additional Latent-DPS step after unconditional sampling can (sometimes) marginally increase the overall performance. We perform an ablation study on the performance of including Latent-DPS in the Appendix.

**Remapping Back to $\boldsymbol{z}_t$.** Following the flowchart in Figure 2, the next step is to map the measurement-consistent sample $\hat{\boldsymbol{z}}_0(\boldsymbol{y})$ back onto the data manifold defined by the noisy samples at time $t$ to continue the reverse sampling process. Doing so would be equivalent to computing the posterior distribution $p(\boldsymbol{z}_t|\boldsymbol{y})$. To incorporate $\hat{\boldsymbol{z}}_0(\boldsymbol{y})$ into the posterior, we propose to construct an auxiliary distribution $p(\hat{\boldsymbol{z}}_t|\hat{\boldsymbol{z}}_0(\boldsymbol{y}), \boldsymbol{y})$ to replace $p(\boldsymbol{z}_t|\boldsymbol{y})$. Here, $\hat{\boldsymbol{z}}_t$ denotes the remapped sample and $\boldsymbol{z}'_t$ denotes the unconditional sample before remapping. One simple way of computing this distribution to obtain $\hat{\boldsymbol{z}}_t$ is shown in Proposition 1.

**Proposition 1** (Stochastic Encoding). *Since the sample $\hat{\boldsymbol{z}}_t$ given $\hat{\boldsymbol{z}}_0(\boldsymbol{y})$ and measurement $\boldsymbol{y}$ is conditionally independent of $\boldsymbol{y}$, we have that*

$$p(\hat{\boldsymbol{z}}_t|\hat{\boldsymbol{z}}_0(\boldsymbol{y}), \boldsymbol{y}) = p(\hat{\boldsymbol{z}}_t|\hat{\boldsymbol{z}}_0(\boldsymbol{y})) = \mathcal{N}(\sqrt{\bar{\alpha}_t}\hat{\boldsymbol{z}}_0(\boldsymbol{y}), (1 - \bar{\alpha}_t)\boldsymbol{I}). \tag{12}$$

We defer all of the proofs to the Appendix. Proposition 1 provides us a way of computing $\hat{\boldsymbol{z}}_t$, which we refer to as *stochastic encoding*. However, we observe that using stochastic encoding can incur a high variance when $t$ is farther away from $t = 0$, where the ground truth signal exists. This large variance can often lead to noisy image reconstructions. To address this issue, we propose a posterior

| Method | Super Resolution $4\times$ | | | Inpainting (Random 70%) | | |
|---|---|---|---|---|---|---|
| | LPIPS↓ | PSNR↑ | SSIM↑ | LPIPS↓ | PSNR↑ | SSIM↑ |
| DPS Chung et al. (2023a) | $0.173 \pm 0.04$ | $28.41 \pm 2.20$ | $0.782 \pm 0.06$ | $\underline{0.102} \pm 0.02$ | $\underline{32.48} \pm 2.30$ | $\underline{0.899} \pm 0.03$ |
| MCG Chung et al. (2022) | $0.193 \pm 0.03$ | $25.92 \pm 2.35$ | $0.740 \pm 0.05$ | $0.134 \pm 0.03$ | $29.53 \pm 2.70$ | $0.847 \pm 0.05$ |
| ADMM-PnP Ahmad et al. (2019) | $0.304 \pm 0.04$ | $21.08 \pm 3.13$ | $0.631 \pm 0.11$ | $0.627 \pm 0.07$ | $15.40 \pm 2.09$ | $0.342 \pm 0.09$ |
| DDRM Kawar et al. (2022) | $0.151 \pm 0.03$ | $\underline{29.49} \pm 1.93$ | $0.817 \pm 0.05$ | $0.166 \pm 0.03$ | $27.69 \pm 1.54$ | $0.798 \pm 0.04$ |
| DMPS Meng & Kabashima (2022) | $\underline{0.147} \pm 0.03$ | $28.48 \pm 1.92$ | $\underline{0.811} \pm 0.05$ | $0.175 \pm 0.03$ | $28.84 \pm 1.65$ | $0.826 \pm 0.03$ |
| Latent-DPS | $0.272 \pm 0.05$ | $26.83 \pm 2.00$ | $0.690 \pm 0.07$ | $0.226 \pm 0.04$ | $26.23 \pm 1.84$ | $0.703 \pm 0.07$ |
| PSLD-LDM* Rout et al. (2023) | $0.209 \pm 0.10$ | $27.61 \pm 2.95$ | $0.704 \pm 0.17$ | $0.260 \pm 0.08$ | $27.07 \pm 2.45$ | $0.689 \pm 0.11$ |
| ReSample (Ours) | $\mathbf{0.144} \pm 0.029$ | $\mathbf{30.45} \pm 2.09$ | $\mathbf{0.832} \pm 0.05$ | $\mathbf{0.082} \pm 0.02$ | $\mathbf{32.77} \pm 2.23$ | $\mathbf{0.903} \pm 0.03$ |

Table 1: **Quantitative results of super resolution and inpainting on the CelebA-HQ dataset.** Input images have an additive Gaussian noise with $\sigma_{\boldsymbol{y}} = 0.01$. Best results are in bold and second best results are underlined.

sampling technique that reduces the variance by additionally conditioning on $\boldsymbol{z}'_t$, the unconditional sample at time $t$. Here, the intuition is that by using information of $\boldsymbol{z}'_t$, we can get closer to the ground truth $\boldsymbol{z}_t$, which effectively reduces the variance. In Lemma 2, under some mild assumptions, we show that this new distribution $p(\hat{\boldsymbol{z}}_t|\boldsymbol{z}'_t, \hat{\boldsymbol{z}}_0(\boldsymbol{y}), \boldsymbol{y})$ is a tractable Gaussian distribution.

**Proposition 2** (Stochastic Resampling). *Suppose that $p(\boldsymbol{z}'_t|\hat{\boldsymbol{z}}_t, \hat{\boldsymbol{z}}_0(\boldsymbol{y}), \boldsymbol{y})$ is normally distributed such that $p(\boldsymbol{z}'_t|\hat{\boldsymbol{z}}_t, \hat{\boldsymbol{z}}_0(\boldsymbol{y}), \boldsymbol{y}) = \mathcal{N}(\boldsymbol{\mu}_t, \sigma_t^2)$. If we let $p(\hat{\boldsymbol{z}}_t|\hat{\boldsymbol{z}}_0(\boldsymbol{y}), \boldsymbol{y})$ be a prior for $\boldsymbol{\mu}_t$, then the posterior distribution $p(\hat{\boldsymbol{z}}_t|\boldsymbol{z}'_t, \hat{\boldsymbol{z}}_0(\boldsymbol{y}), \boldsymbol{y})$ is given by*

$$p(\hat{\boldsymbol{z}}_t|\boldsymbol{z}'_t, \hat{\boldsymbol{z}}_0(\boldsymbol{y}), \boldsymbol{y}) = \mathcal{N}\left(\frac{\sigma_t^2\sqrt{\bar{\alpha}_t}\hat{\boldsymbol{z}}_0(\boldsymbol{y}) + (1-\bar{\alpha}_t)\boldsymbol{z}'_t}{\sigma_t^2 + (1-\bar{\alpha}_t)}, \frac{\sigma_t^2(1-\bar{\alpha}_t)}{\sigma_t^2 + (1-\bar{\alpha}_t)}\boldsymbol{I}\right). \qquad (13)$$

We refer to this new mapping technique as *stochastic resampling*. Since we do not have access to $\sigma_t^2$, it serves as a hyperparameter that we tune in our algorithm. The choice of $\sigma_t^2$ plays a role of controlling the tradeoff between prior consistency and data consistency. If $\sigma_t^2 \to 0$, then we recover unconditional sampling, and if $\sigma_t^2 \to \infty$, we recover stochastic encoding. We observe that this new technique also has several desirable properties, for which we rigorously prove in the next section.

### 3.2 THEORETICAL RESULTS

In Section 3.1, we discussed that stochastic resampling induces less variance than stochastic encoding. Here, we aim to rigorously prove the validity of this statement.

**Lemma 1.** *Let $\tilde{\boldsymbol{z}}_t$ and $\hat{\boldsymbol{z}}_t$ denote the stochastically encoded and resampled image of $\hat{\boldsymbol{z}}_0(\boldsymbol{y})$, respectively. If $VAR(\boldsymbol{z}'_t) > 0$, then we have that $VAR(\hat{\boldsymbol{z}}_t) < VAR(\tilde{\boldsymbol{z}}_t)$.*

**Theorem 1.** *If $\hat{\boldsymbol{z}}_0(\boldsymbol{y})$ is measurement-consistent such that $\boldsymbol{y} = \mathcal{A}(\mathcal{D}(\hat{\boldsymbol{z}}_0(\boldsymbol{y})))$, i.e. $\hat{\boldsymbol{z}}_0 = \hat{\boldsymbol{z}}_0(\boldsymbol{z}_{t+1}) = \hat{\boldsymbol{z}}_0(\boldsymbol{y})$, then stochastic resample is unbiased such that $\mathbb{E}[\hat{\boldsymbol{z}}_t|\boldsymbol{y}] = \mathbb{E}[\boldsymbol{z}'_t]$.*

These two results, Lemma 1 and Theorem 1, prove the benefits of stochastic resampling. At a high-level, these proofs rely on the fact the posterior distributions of both stochastic encoding and resampling are Gaussian and compare their respective means and variances. In the following result, we characterize the variance induced by stochastic resampling, and show that as $t \to 0$, the variance decreases, giving us a reconstructed image that is of better quality.

**Theorem 2.** *Let $\boldsymbol{z}_0$ denote a sample from the data distribution and $\boldsymbol{z}_t$ be a sample from the noisy perturbed distribution at time $t$. Then,*

$$Cov(\boldsymbol{z}_0|\boldsymbol{z}_t) = \frac{(1-\bar{\alpha}_t)^2}{\bar{\alpha}_t}\nabla^2_{\boldsymbol{z}_t}\log p_{\boldsymbol{z}_t}(\boldsymbol{z}_t) + \frac{1-\bar{\alpha}_t}{\bar{\alpha}_t}\boldsymbol{I}.$$

By Theorem 2, notice that since as $\alpha_t$ is an increasing sequence that converges to 1 as $t$ decreases, the variance between the ground truth $\boldsymbol{z}_0$ and the estimated $\hat{\boldsymbol{z}}_0$ decreases to 0 as $t \to 0$, assuming that $\nabla^2_{\boldsymbol{z}_t}\log p_{\boldsymbol{z}_t}(\boldsymbol{z}_t) < \infty$. Following our theory, we empirically show that stochastic resampling can reconstruct signals that are less noisy than stochastic encoding, as shown in the next section.

### 4 EXPERIMENTS

We conduct experiments to solve both linear and nonlinear inverse problems on natural and medical images. We compare our algorithm to several state-of-the-art methods that directly apply the diffusion models that are trained in the pixel space: DPS (Chung et al., 2023a), Manifold Constrained

| Method | Nonlinear Deblurring | | | Gaussian Deblurring | | |
|---|---|---|---|---|---|---|
| | LPIPS↓ | PSNR↑ | SSIM↑ | LPIPS↓ | PSNR↑ | SSIM↑ |
| DPS Chung et al. (2023a) | 0.230 ± 0.065 | 26.81 ± 2.84 | 0.720 ± 0.077 | 0.175 ± 0.03 | 28.36 ± 2.12 | 0.772 ± 0.07 |
| MCG Chung et al. (2022) | - | - | - | 0.517 ± 0.06 | 15.85 ± 1.08 | 0.536 ± 0.08 |
| ADMM-PnP Ahmad et al. (2019) | 0.499 ± 0.073 | 16.17 ± 4.01 | 0.359 ± 0.140 | 0.289 ± 0.04 | 20.98 ± 4.51 | 0.602 ± 0.15 |
| DDRM Kawar et al. (2022) | - | - | - | 0.193 ± 0.04 | 26.88 ± 1.96 | 0.747 ± 0.07 |
| DMPS Meng & Kabashima (2022) | - | - | - | 0.206 ± 0.04 | 26.45 ± 1.83 | 0.726 ± 0.07 |
| Latent-DPS | 0.225 ± 0.04 | 26.18 ± 1.73 | 0.703 ± 0.07 | 0.205 ± 0.04 | 27.42 ± 1.84 | 0.729 ± 0.07 |
| PSLD-LDM* Rout et al. (2023) | - | - | - | 0.323 ± 0.09 | 24.21 ± 2.79 | 0.548 ± 0.12 |
| ReSample (Ours) | **0.153** ± 0.03 | **30.18** ± 2.21 | **0.828** ± 0.05 | **0.148** ± 0.04 | **30.69** ± 2.14 | **0.832** ± 0.05 |

Table 2: **Quantitative results of Gaussian and nonlinear deblurring on the CelebA-HQ dataset.** Input images have an additive Gaussian noise with $\sigma_y = 0.01$. Best results are in bold and second best results are underlined. For nonlinear deblurring, some baselines are omitted, as they can only solve *linear* inverse problems.

| Method | Abdominal | | Head | | Chest | |
|---|---|---|---|---|---|---|
| | PSNR↑ | SSIM↑ | PSNR↑ | SSIM↑ | PSNR↑ | SSIM↑ |
| Latent-DPS | 26.80 ±1.09 | 0.870 ±0.026 | 28.64 ±5.38 | 0.893 ±0.058 | 25.67±1.14 | 0.822 ±0.033 |
| MCG (Chung et al., 2022) | 29.41 ±3.14 | 0.857 ±0.041 | 28.28 ±3.08 | 0.795 ±0.116 | 27.92 ±2.48 | 0.842 ±0.036 |
| DPS (Chung et al., 2023a) | 27.33 ±2.68 | 0.715 ±0.031 | 24.51 ±2.77 | 0.665 ±0.058 | 24.73 ±1.84 | 0.682 ±0.113 |
| PnP-UNet (Gilton et al., 2021) | 32.84 ±1.29 | 0.942 ±0.008 | 33.45 ±3.25 | 0.945 ±0.023 | 29.67 ±1.14 | 0.891 ±0.011 |
| FBP | 26.29 ±1.24 | 0.727 ±0.036 | 26.71 ±5.02 | 0.725 ±0.106 | 24.12 ±1.14 | 0.655 ±0.033 |
| FBP-UNet (Jin et al., 2017) | 32.77 ±1.21 | 0.937 ±0.013 | 31.95 ±3.32 | 0.917 ±0.048 | 29.78 ±1.12 | 0.885 ±0.016 |
| ReSample (Ours) | **35.91** ±1.22 | **0.965** ±0.007 | **37.82** ±5.31 | **0.978** ±0.014 | **31.72** ±0.912 | **0.922** ±0.011 |

Table 3: **Quantitative results of CT reconstruction on the LDCT dataset**. Best results are in bold and second best results are underlined.

Gradients (MCG) (Chung et al., 2022), Denoising Diffusion Destoration Models (DDRM) (Kawar et al., 2022), Diffusion Model Posterior Sampling (DMPS) (Meng & Kabashima, 2022). Then we compare an algorithm that uses a plug-and-play approach that apply a pretrained deep denoiser for inverse problems (ADMM-PnP) (Ahmad et al., 2019). We also compare our algorithm to Latent-DPS and Posterior Sampling with Latent Diffusion (PSLD) (Rout et al., 2023), a concurrent work we recently notice also tackling latent diffusion models. Various quantitative metrics are used for evaluation including Learned Perceptual Image Patch Similarity (LPIPS) distance, peak signal-to-noise-ratio (PSNR), and structural similarity index (SSIM). Lastly, we conduct ablation study to compare the performance of stochastic encoding and our proposed stochastic resampling technique as mentioned in Section 3.1, and also demonstrate the memory efficiency gained by leveraging LDMs.

**Experiments on Natural Images.** For the experiments on natural images, we use datasets FFHQ (Kazemi & Sullivan, 2014), CelebA-HQ (Liu et al., 2015), and LSUN-Bedroom (Yu et al., 2016) with the image resolution of $256 \times 256 \times 3$. We take pre-trained latent diffusion models LDM-VQ4 trained on FFHQ and CelebA-HQ provided by (Rombach et al., 2022) with autoencoders that yield images of size $64 \times 64 \times 3$, and DDPMs (Ho et al., 2020) also trained on FFHQ and CelebA-HQ training sets. Then, we sample 100 images from both the FFHQ and CelebA-HQ validation sets for testing evaluation. For computing quantitative results, all images are normalized to the range $[0, 1]$. All experiments had Gaussian measurement noise with standard deviation $\sigma_y = 0.01$. Due to limited space, we put the results on FFHQ and details of the hyperparameters to the Appendix.

For linear inverse problems, we consider the following tasks: (1) Gaussian deblurring, (2) inpainting (with a random mask), and (3) super resolution. For Gaussian deblurring, we use a kernel with size $61 \times 61$ with standard deviation 3.0. For super resolution, we use bicubic downsampling and a random mask with varying levels of missing pixels for inpainting. For nonlinear inverse problems, we consider nonlinear deblurring as proposed by (Chung et al., 2023a). The quantitative results are displayed in Tables 1 and 2, with qualitative results in Figure 3. In Tables 1 and 2, we can see that ReSample significantly outperforms all of the baselines across all three metrics on the CelebA-HQ dataset. We also observe that ReSample performs better than or comparable to all baselines on the FFHQ dataset as shown in the Appendix. Remarkably, our method excels in handling nonlinear inverse problems, further demonstrating the flexibility of our algorithm. We further demonstrate the superiority of ReSample for handling nonlinear inverse problems in Figure 6a, where we show that we can consistently outperform DPS.

---

*We have updated the baseline results for PSLD. More details are provided in the Appendix (Section A.4).

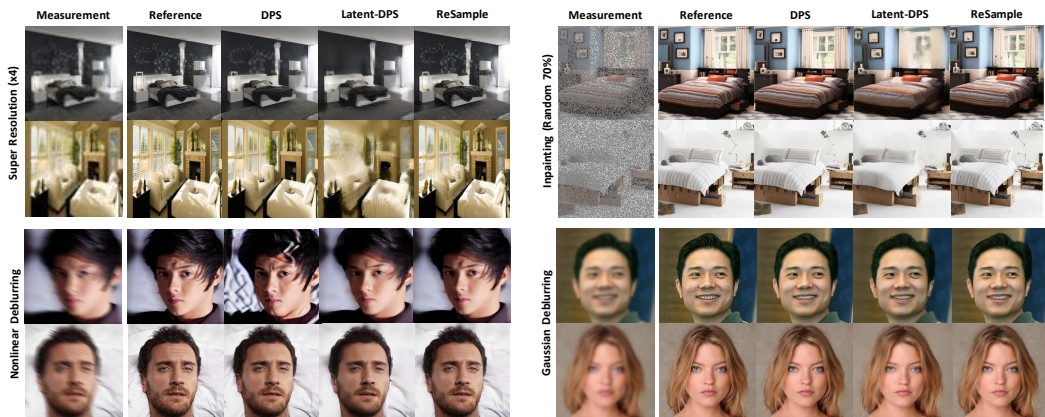

Figure 3: **Qualitative results of multiple tasks on the LSUN-Bedroom and CelebA-HQ datasets.** All inverse problems have Gaussian measurement noise with variance $\sigma_{\boldsymbol{y}} = 0.01$.

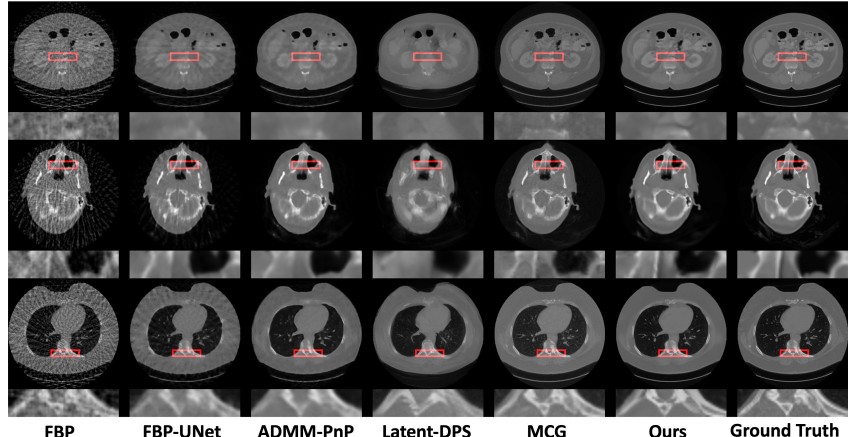

Figure 4: **Qualitative results of CT reconstruction on the LDCT dataset.** We annotate the critical image structures in a red box, and zoom in below the image.

**Experiments on Medical Images.** For experiments on medical images, we fine-tune LDMs on 2000 2D computed tomography (CT) images with image resolution of $256 \times 256$, randomly sampled from the AAPM LDCT dataset (Moen et al., 2021) of 40 patients, and test on the 300 2D CT images from the remaining 10 patients. We take the FFHQ LDM-VQ4 model provided by Rombach et al. (2022) as the pre-trained LDM followed by 100K fine-tuning iterations. Then, we simulate CT measurements (sinograms) with a parallel-beam geometry using 25 projection angles equally distributed across 180 degrees using the `torch-radon` package (Ronchetti, 2020). Following the natural images, the sinograms were perturbed with additive Gaussian measurement noise with $\sigma_{\boldsymbol{y}} = 0.01$. Along with the baselines used in the natural image experiments, we also compare to the following methods: (1) Filtered Backprojection (FBP), which is a standard CT reconstruction technique, and (2) FBP-UNet, a supervised method that uses a UNet model as its backbone. For FBP-Unet and PnP-UNet, we trained a model on 3480 2D CT from the training set. For MCG and DPS, we used the pre-trained checkpoints provided by Chung et al. (2022). We present visual reconstructions with highlighted critical anatomic structures in Figure 4 and quantitative results in Table 3. Here, we observe that our algorithm outperforms all of the baselines in terms of PSNR and SSIM. Visually, we also observe that our algorithm is able to reconstruct smoother images that have more accurate and sharper details.

**Effectiveness of the Resampling Technique.** Here, we validate our theoretical results by conducting ablation studies on stochastic resampling. Specifically, we perform experiments on the LSUN-Bedroom and CelebA-HQ datasets with tasks of Gaussian deblurring and super-resolution. As shown in Figure 5, we observe that stochastic resampling reconstructs smoother images with higher PSNRs compared to stochastic encoding, corroborating our theory.

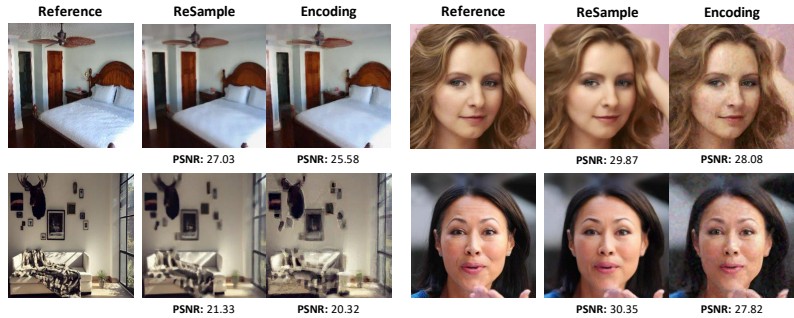

Figure 5: **Effectiveness of our resampling technique compared to stochastic encoding.** Results were demonstrated on the LSUN-Bedroom and CelebA-HQ datasets with measurement noise of $\sigma_{\boldsymbol{y}} = 0.05$ to highlight the effectiveness of stochastic resample.

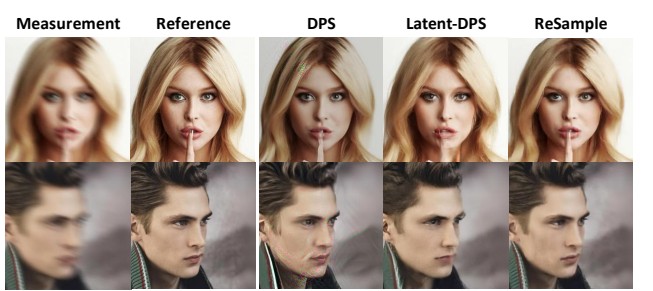

(a) More examples of nonlinear deblurring

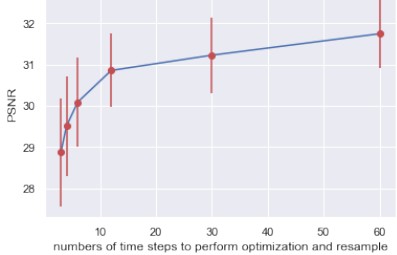

(b) Performance v.s. number of time steps to perform resample

Figure 6: Left: Additional results on nonlinear deblurring highlighting the performance of ReSample. Right: Ablation study on the ReSample frequency on the performance.

**Effectiveness of Hard Data Consistency.** In Figure 6b, we perform an ablation study on the ReSample frequency on CT reconstruction. This observation is in line with what we expect intuitively, as more ReSample time steps (i.e., more data consistency) lead to more accurate reconstructions.

**Memory Efficiency.** To demonstrate memory efficiency, we use the command `nvidia-smi` to monitor the memory consumption during solving an inverse problem. We present the memory usage for Gaussian deblurring on the FFHQ dataset in Table 4. Although the entire LDM models occupy more memory due to the autoencoders, our algorithm itself exhibits memory efficiency, resulting in lower overall memory usage. This highlights its potential in domains like medical imaging, where memory plays a crucial role in feasibility.

Table 4: Memory Usage of Different Methods for Gaussian Deblurring on the FFHQ Dataset

| Model | Algorithm | Model Only | Memory Increment | Total |
|-------|-----------|------------|------------------|-------|
| DDPM | DPS | **1953MB** | +3416MB (175%) | 5369MB |
| | MCG | | +3421MB (175%) | 5374MB |
| | DMPS | | +5215MB (267 %) | 7168MB |
| | DDRM | | +18833MB (964 %) | 20786MB |
| LDM | PSLD | 3969MB | +5516MB (140%) | 9485MB |
| | ReSample | | **+1040MB (26.2%)** | **5009MB** |

## 5 CONCLUSION

In this paper, we propose *ReSample*, an algorithm that can effectively leverage LDMs to solve *general* inverse problems. We demonstrated that our algorithm can reconstruct high-quality images compared to many baselines, including those in the pixel space. One limitation of our method lies in the computational overhead of hard data consistency, which we leave as a significant challenge for future work to address and improve upon.

## 6 REPRODUCIBILITY STATEMENT

To ensure the reproducibility of our results, we thoroughly detail the hyperparameters employed in our algorithm in the Appendix. Additionally, we provide a comprehensive explanation of the configuration of all baselines used in our experiments. As we use pre-trained diffusion models throughout our experiments, they are readily accessible online. Lastly, our code is available at `https://github.com/soominkwon/resample`.

## 7 ACKNOWLEDGEMENTS

BS and LS acknowledges support from U-M MIDAS PODS Grant and U-M MICDE Catalyst Grant, and computing resource support from NSF ACCESS Program and Google Cloud Research Credits Program. This work used NCSA Delta GPU through allocation CIS230133 and ELE230011 from the Advanced Cyberinfrastructure Coordination Ecosystem: Services & Support (ACCESS) program, which is supported by National Science Foundation grants #2138259, #2138286, #2138307, #2137603, and #2138296. SMK and QQ acknowledge support from U-M START & PODS grants, NSF CAREER CCF-2143904, NSF CCF-2212066, NSF CCF-2212326, ONR N00014-22-1-2529, AWS AI Award, and a gift grant from KLA.

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

# Appendix

In this section, we present additional experimental results to supplement those presented in the main paper. In Section A, we present additional qualitative and quantitative results to highlight the performance of ReSample. In Section B, we briefly discuss some of the implementation details regarding hard data consistency. In Section C, we outline all of the hyperparameters used to produce our results. In Section D, we discuss some more reasons to why Latent-DPS fails to consistently accurately recover the underlying image. Lastly, in Section F, we present our deferred proofs.

## A  ADDITIONAL RESULTS

### A.1  RESULTS ON FFHQ

Here, we provide additional quantitative results on the FFHQ dataset. Similar to the results in the main text on the CelebA-HQ dataset, ReSample outperforms the baselines across many different tasks.

| Method | Super Resolution $4\times$ | | | Inpainting (Random 70%) | | |
|---|---|---|---|---|---|---|
| | LPIPS↓ | PSNR↑ | SSIM↑ | LPIPS↓ | PSNR↑ | SSIM↑ |
| DPS | $\underline{0.175} \pm 0.04$ | $\underline{28.47} \pm 2.09$ | $\underline{0.793} \pm 0.05$ | $\underline{0.106} \pm 0.03$ | $\mathbf{32.32} \pm 2.21$ | $\mathbf{0.897} \pm 0.04$ |
| MCG | $0.223 \pm 0.04$ | $23.74 \pm 3.18$ | $0.673 \pm 0.07$ | $0.178 \pm 0.03$ | $24.89 \pm 2.57$ | $0.731 \pm 0.06$ |
| ADMM-PnP | $0.303 \pm 0.03$ | $21.30 \pm 3.98$ | $0.760 \pm 0.04$ | $0.308 \pm 0.07$ | $15.87 \pm 2.09$ | $0.608 \pm 0.09$ |
| DDRM | $0.257 \pm 0.03$ | $27.51 \pm 1.79$ | $0.753 \pm 0.05$ | $0.287 \pm 0.03$ | $24.97 \pm 1.51$ | $0.680 \pm 0.05$ |
| DMPS | $0.181 \pm 0.03$ | $27.21 \pm 1.92$ | $0.766 \pm 0.06$ | $0.150 \pm 0.03$ | $28.17 \pm 1.58$ | $0.814 \pm 0.04$ |
| Latent-DPS | $0.344 \pm 0.05$ | $24.65 \pm 1.77$ | $0.609 \pm 0.07$ | $0.270 \pm 0.05$ | $27.08 \pm 1.97$ | $0.727 \pm 0.06$ |
| PSLD | $0.580 \pm 0.05$ | $15.81 \pm 1.54$ | $0.183 \pm 0.07$ | $0.270 \pm 0.06$ | $25.61 \pm 2.02$ | $0.630 \pm 0.09$ |
| ReSample (Ours) | $\mathbf{0.164} \pm 0.03$ | $\mathbf{28.90} \pm 1.86$ | $\mathbf{0.804} \pm 0.05$ | $\mathbf{0.099} \pm 0.02$ | $\underline{31.34} \pm 2.11$ | $\underline{0.890} \pm 0.03$ |

Table 5: Comparison of quantitative results on inverse problems on the FFHQ dataset. Input images have an additive Gaussian noise with $\sigma_y = 0.01$. Best results are in bold and second best results are underlined.

| Method | Nonlinear Deblurring | | | Gaussian Deblurring | | |
|---|---|---|---|---|---|---|
| | LPIPS↓ | PSNR↑ | SSIM↑ | LPIPS↓ | PSNR↑ | SSIM↑ |
| DPS | $\underline{0.197} \pm 0.06$ | $\underline{27.13} \pm 3.11$ | $\underline{0.762} \pm 0.08$ | $\mathbf{0.169} \pm 0.04$ | $\underline{27.70} \pm 2.04$ | $\underline{0.774} \pm 0.05$ |
| MCG | - | - | - | $0.371 \pm 0.04$ | $25.33 \pm 1.74$ | $0.668 \pm 0.06$ |
| ADMM-PnP | $0.424 \pm 0.03$ | $21.80 \pm 1.42$ | $0.497 \pm 0.05$ | $0.399 \pm 0.03$ | $21.23 \pm 3.01$ | $0.675 \pm 0.06$ |
| DDRM | - | - | - | $0.299 \pm 0.039$ | $26.51 \pm 1.67$ | $0.702 \pm 0.06$ |
| DMPS | - | - | - | $0.227 \pm 0.036$ | $26.04 \pm 1.75$ | $0.699 \pm 0.066$ |
| Latent-DPS | $0.319 \pm 0.05$ | $24.52 \pm 1.75$ | $0.637 \pm 0.07$ | $0.258 \pm 0.05$ | $25.98 \pm 1.66$ | $0.704 \pm 0.06$ |
| PSLD | - | - | - | $0.422 \pm 0.11$ | $20.08 \pm 2.97$ | $0.400 \pm 0.16$ |
| ReSample (Ours) | $\mathbf{0.171} \pm 0.04$ | $\mathbf{30.03} \pm 1.78$ | $\mathbf{0.838} \pm 0.04$ | $\underline{0.201} \pm 0.04$ | $\mathbf{28.73} \pm 1.87$ | $\mathbf{0.794} \pm 0.05$ |

Table 6: Comparison of quantitative results on inverse problems on the FFHQ dataset. Input images have an additive Gaussian noise with $\sigma_y = 0.01$. Best results are in bold and second best results are underlined.

We observe that ReSample outperforms all baselines in nonlinear deblurring and super resolution, while demonstrating comparable performance to DPS (Chung et al., 2023a) in Gaussian deblurring and inpainting. An interesting observation is that ReSample exhibits the largest performance gap from DPS in nonlinear deblurring and the smallest performance gap in random inpainting, mirroring the pattern we observed in the results of the CelebA-HQ dataset. This may suggest that ReSample performs better than baselines when the forward operator is more complex.

### A.2  ADDITIONAL RESULTS ON BOX INPAINTING

In this section, we present both qualitative and quantitative results on a more challenging image inpainting setting. More specifically, we consider the "box" inpainting setting, where the goal is to

recover a ground truth image in which large regions of pixels are missing. We present our results in Figure 7 and Table 7, where we compare the performance of our algorithm to PSLD (Rout et al., 2023) (latent-space algorithm) and DPS (Chung et al., 2023a) (pixel-space algorithm). Even in this challenging setting, we observe that ReSample can outperform the baselines, highlighting the effectiveness of our method.

| Method | Box Inpainting | | |
|---|---|---|---|
| | LPIPS↓ | PSNR↑ | SSIM↑ |
| DPS | 0.127 | 22.85 | 0.861 |
| DDRM | 0.120 | 24.33 | 0.860 |
| DMPS | 0.233 | 22.01 | 0.803 |
| Latent-DPS | 0.199 | 23.14 | 0.784 |
| PSLD | 0.201 | 23.99 | 0.787 |
| ReSample (Ours) | **0.093** | **24.67** | **0.892** |

Table 7: Comparison of quantitative results for box inpainting on CelebA-HQ dataset. Input images have an additive Gaussian noise with $\sigma_y = 0.01$. Best results are in bold and second best results are underlined.

## A.3 ADDITIONAL PRELIMINARY RESULTS ON HIGH-RESOLUTION IMAGES

In this section, we present techniques for extending our algorithm to address inverse problems with high-resolution images. One straightforward method involves using arbitrary-resolution random noise as the initial input for the LDM. The convolutional architecture shared by the UNet in both the LDM and the autoencoder enables the generation of images at arbitrary resolutions, as demonstrated by Rombach et al. (2022). To illustrate the validity of this approach, we conducted a random-inpainting experiment with images of dimensions $(512 \times 512 \times 3)$ and report the results in Figure 8. In Figure 8, we observe that our method can achieve excellent performance even in this high-resolution setting. Other possible methods include obtaining an accurate pair of encoders and decoders that can effectively map higher-resolution images to the low-dimensional space, and then running our algorithm using the latent diffusion model. We believe that these preliminary results represent an important step towards using LDMs as generative priors for solving inverse problems for high-resolution images.

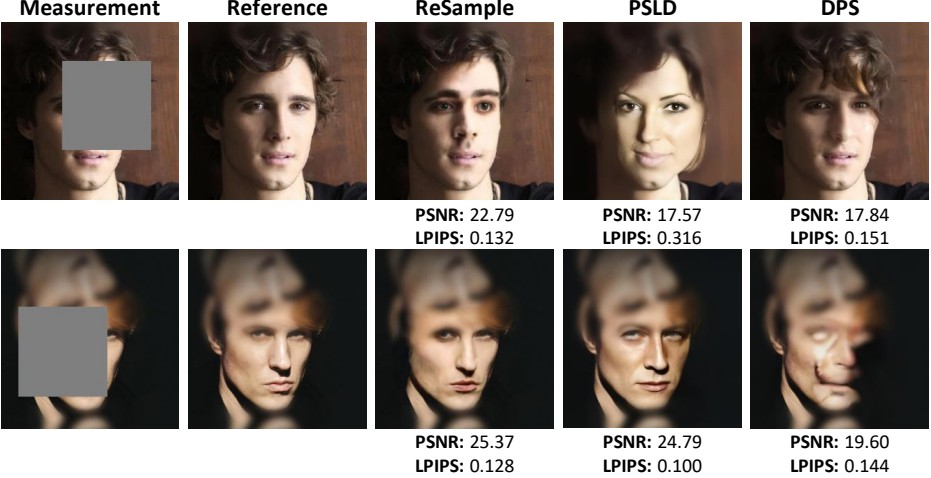

Figure 7: Qualitative results on box inpainting with measurement noise $\sigma_y = 0.01$. These results highlight the effectiveness of ReSample on more difficult inverse problem tasks.

## A.4 DISCUSSION ON PSLD BASELINE

To better reflect the performance of PSLD (Rout et al., 2023), we re-ran several experiments with PSLD while collaborating with the original authors to fine-tune hyperparameters. Furthermore,

| Measurement | Reference | ReSample |

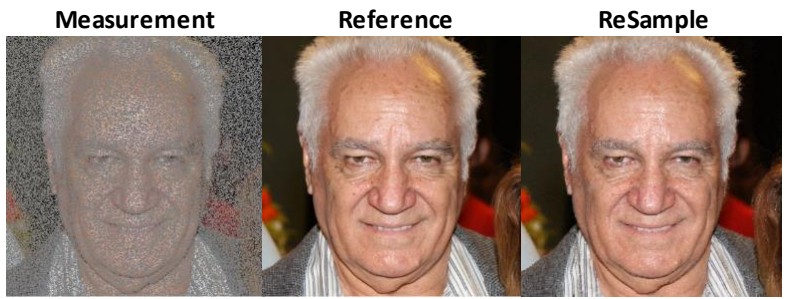

Figure 8: Additional results on 70% random inpainting with high-resolution images ($512 \times 512 \times 3$) with measurement noise $\sigma_{\boldsymbol{y}} = 0.01$.

as the work by Rout et al. (2023) initially employed the stable diffusion (SD) model rather than the LDM model, we have relabeled the baselines as "PSLD-LDM". We report the hyperparameters used to generate the new results, as well as the ones used to generate the previous results in Table 8. The previous hyperparameters were chosen based on the implementation details provided by Rout et al. (2023) in Section B.1, where they stated that they used $\gamma = 0.1$ and that the hyperparameters were available in the codebase. However, it has been brought to our attention that this was not the optimal hyperparameter. Therefore, we additionally tuned the parameter so that each baseline could obtain the best results. We have also conducted additional experiments directly comparing to PSLD-LDM for inpainting tasks on the FFHQ 1K validation set and present the results in Table 9. Throughout these results, we still observe that our algorithm largely outperforms the baselines, including PSLD.

| Hyperparameter (Rout et al., 2023) | SR (FFHQ) | SR (CelebA) | Gaussian Deblur (CelebA) |
|---|---|---|---|
| Previous ($\gamma$) | 0.1 | 0.1 | 0.1 |
| New ($\gamma$) | 1.0 | 1.1 | 0.15 |

Table 8: Different hyperparameters used to generate PSLD results. SR refers to super resolution ($4\times$), "previous" and "new" refer to the hyperparameter $\gamma$ used to generate the old and new results, respectively.

| Method | Inpainting (Random) | | | Inpainting (Box) | | |
|---|---|---|---|---|---|---|
| | LPIPS↓ | PSNR↑ | SSIM↑ | LPIPS↓ | PSNR↑ | SSIM↑ |
| DPS | 0.212 | 29.49 | 0.844 | 0.214 | 23.39 | 0.798 |
| PSLD-LDM | 0.221 | 30.31 | 0.851 | 0.158 | **24.22** | 0.819 |
| ReSample (Ours) | **0.135** | **31.64** | **0.906** | **0.128** | 23.81 | **0.868** |

Table 9: Comparison of quantitative results for different inpainting tasks on the FFHQ 1k validation set. Best results are in bold and second best results are underlined.

## A.5 Additional Qualitative Results

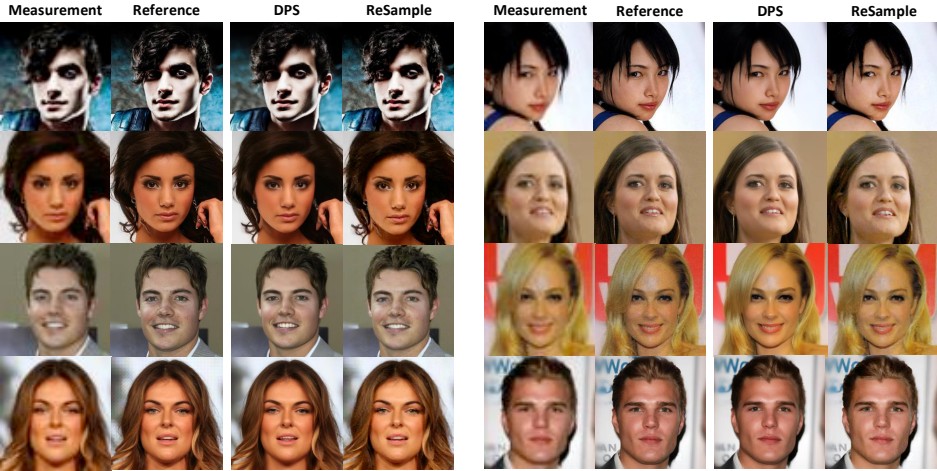

Figure 9: Additional results on super resolution ($4\times$) on the CelebA-HQ dataset with Gaussian measurement noise of variance $\sigma_y = 0.01$.

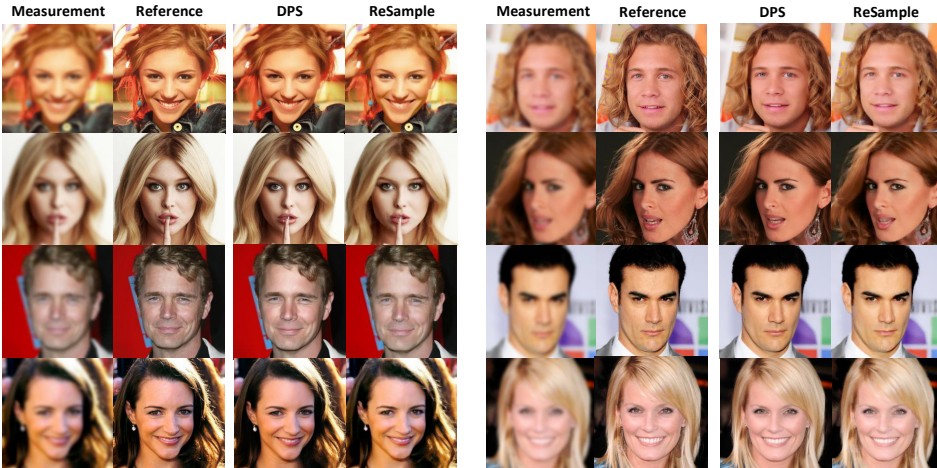

Figure 10: Additional results on Gaussian deblurring on the CelebA-HQ dataset with Gaussian measurement noise of variance $\sigma_y = 0.01$.

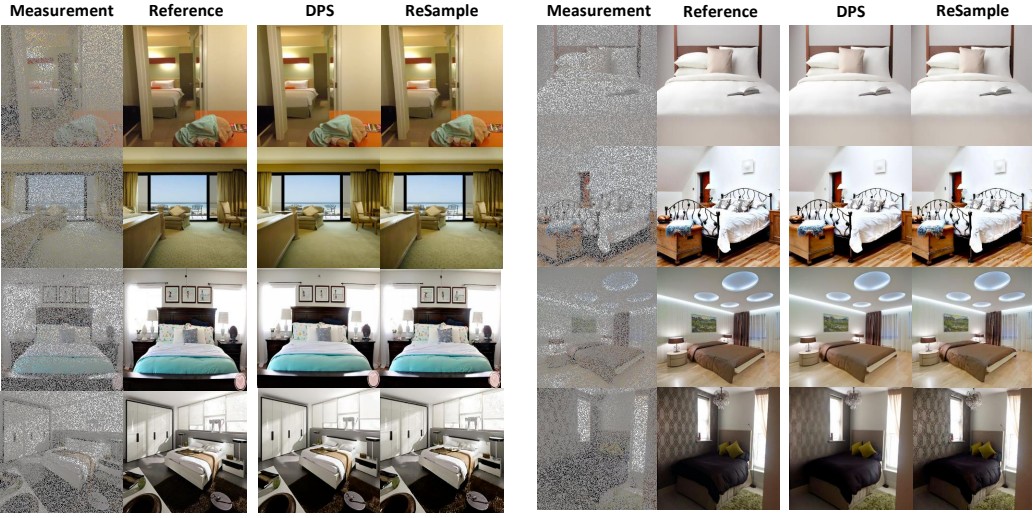

Figure 11: Additional results on inpainting with a random mask (70%) on the LSUN-Bedroom dataset with Gaussian measurement noise of variance $\sigma_{\boldsymbol{y}} = 0.01$.

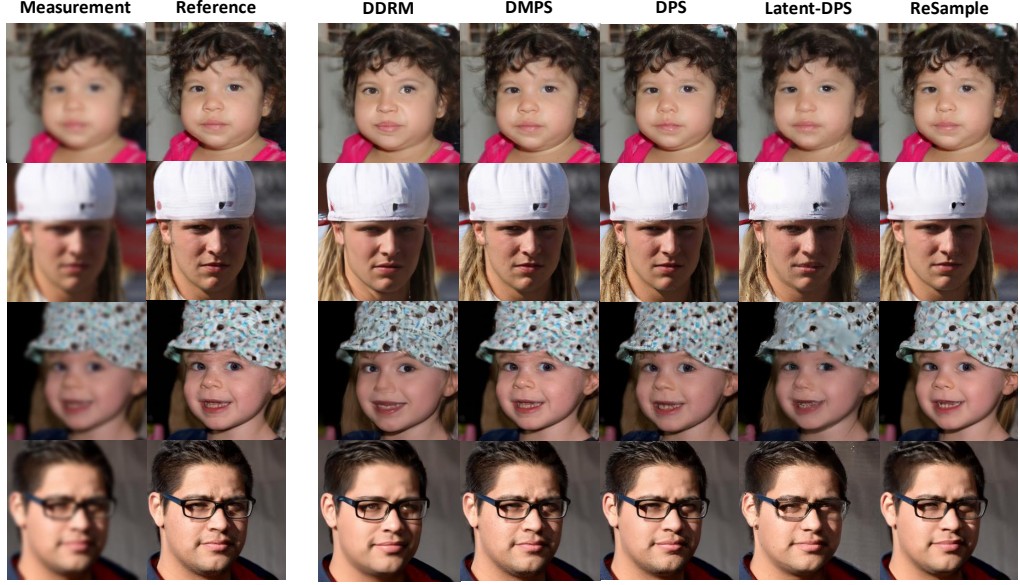

Figure 12: Comparison of algorithms on Gaussian deblurring on the FFHQ dataset with Gaussian measurement noise of variance $\sigma_{\boldsymbol{y}} = 0.01$.

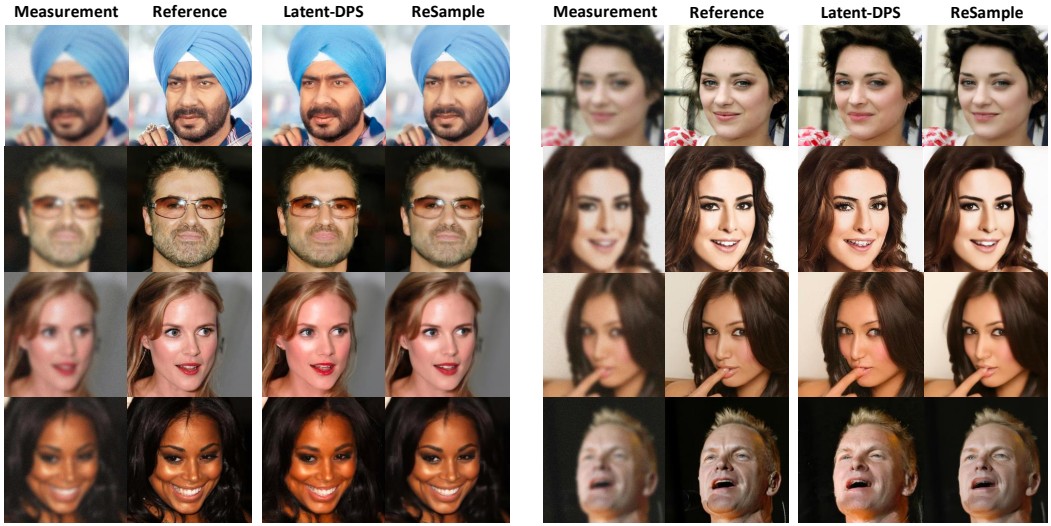

Figure 13: Additional results on Gaussian deblurring with additive Gaussian noise $\sigma_{\boldsymbol{y}} = 0.05$.

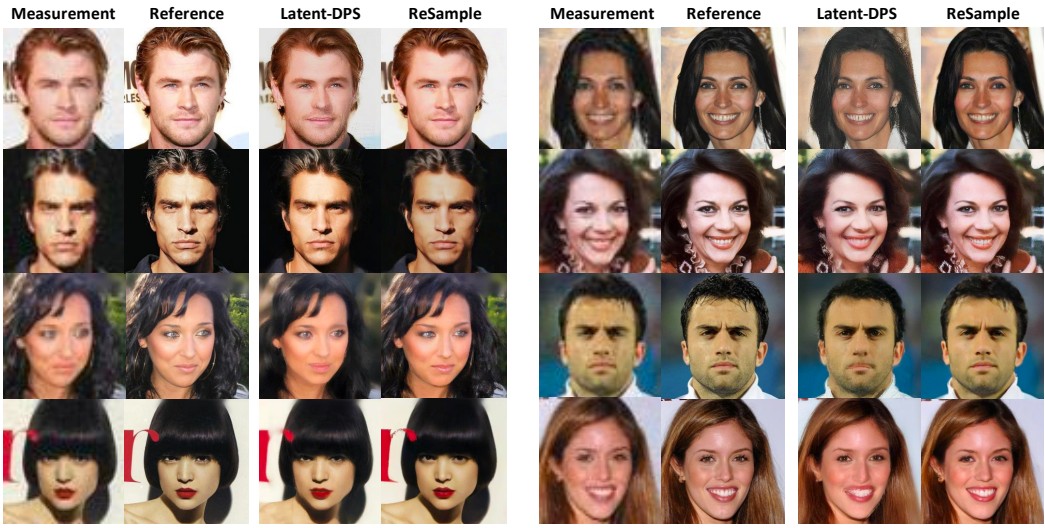

Figure 14: Additional results on super resolution $4\times$ with additive Gaussian noise $\sigma_{\boldsymbol{y}} = 0.05$.

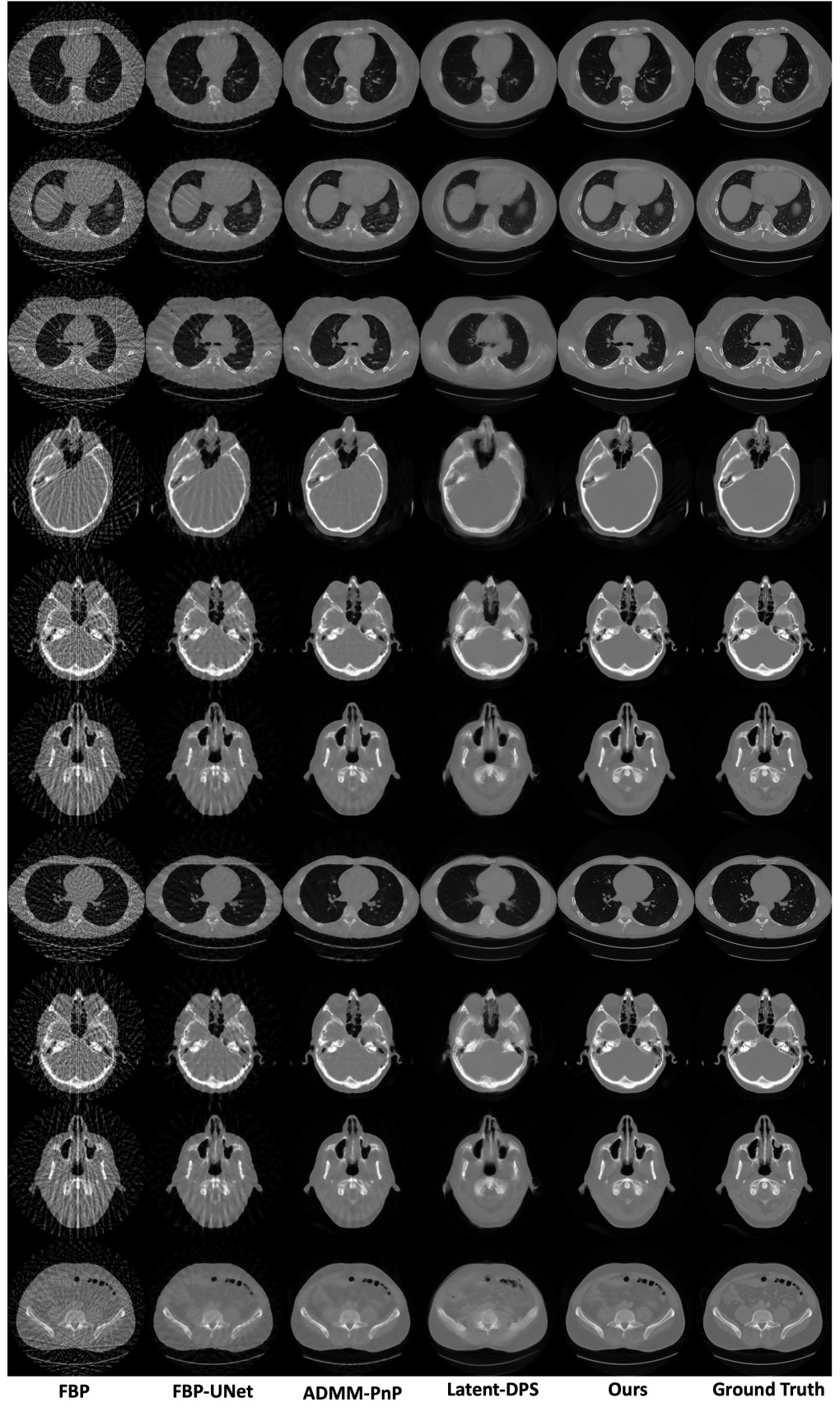

FBP     FBP-UNet     ADMM-PnP     Latent-DPS     Ours     Ground Truth

Figure 15: Additional results on CT reconstruction with additive Gaussian noise $\sigma_{\boldsymbol{y}} = 0.01$.

## A.6   ABLATION STUDIES

Here, we present some ablation studies regarding computational efficiency and stochastic resampling, amongst others.

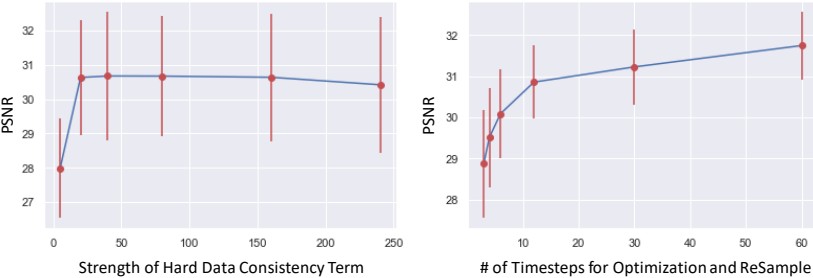

Figure 16: An ablation study on a few of the hyperparameters associated with ReSample. Left: Performance of different values of $\gamma$ (hyperparameter in stochastic resampling) on the CelebA-HQ dataset. Right: CT reconstruction performance as a function of the number of timesteps to perform optimization.

**Effectiveness of Stochastic Resampling.**   Recall that in stochastic resampling, we have one hyperparameter $\sigma_t^2$. In Section C, we discuss that our choice for this hyperparameter is

$$\sigma_t^2 = \gamma \left( \frac{1 - \bar{\alpha}_{t-1}}{\bar{\alpha}_t} \right) \left( 1 - \frac{\bar{\alpha}_t}{\bar{\alpha}_{t-1}} \right),$$

where $\bar{\alpha}$ is an adaptive parameter associated to the diffusion model process. Thus, the only parameter that we need to choose here is $\gamma$. To this end, we perform a study on how $\gamma$ affects the image reconstruction quality. The $\gamma$ term can be interpreted as a parameter that balances the prior consistency with the measurement consistency. In Figure 16 (left) observe that performance increases a lot when $\gamma$ increases from a small value, but plateaus afterwards. We also observe that the larger $\gamma$ gives more fine details on the images, but may introduce additional noise. In Figure 17, we provide visual representations corresponding to the choices of $\gamma$ in order.

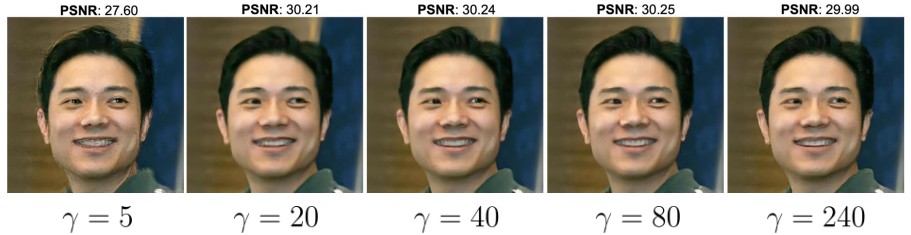

Figure 17: Visual representation of varying $\gamma$ on the CelebA-HQ dataset.

**Effect of the Skip Step Size.**   In this section, we conduct an ablation study to investigate the impact of the skip step size in our algorithm. The skip step size denotes the frequency at which we 'skip' before applying the next hard data consistency. For instance, a skip step size of 10 implies that we apply hard data consistency every 10 iterations of the reverse sampling process. Intuitively, one might expect that fewer skip steps would lead to better reconstructions, indicating more frequent hard data consistency steps. To validate this intuition, we conducted an experiment on the effect of the skip step size on CT reconstruction, and the results are displayed in Figure 18, with corresponding inference times in Table 16. We observe that for skip step sizes ranging from 1 to 10, the results are very similar. However, considering the significantly reduced time required for a skip step size

of 10 as shown in Table 16, we chose a skip step size of 10 in our experimental setting in order to balance the trade-off between reconstruction quality and inference time. Lastly, we would like to note that in Figure 18, a skip step size of 4 exhibits a (very) slight improvement over a skip step of 1. Due to the reverse sampling process initiating with a *random* noise vector, there is a minor variability in the results, with the average outcomes for skip step sizes of 1 and 4 being very similar.

| Algorithm | Skip Step Size | Inference Time | PSNR | SSIM |
|---|---|---|---|---|
| ReSample (Ours) | 1 | 31:20 | 31.74 | 0.922 |
| | 4 | 8:24 | 31.74 | 0.923 |
| | 10 | 3:52 | 31.72 | 0.922 |
| | 20 | 2:16 | 31.23 | 0.918 |
| | 50 | 1:20 | 30.86 | 0.915 |
| | 100 | 1:08 | 30.09 | 0.906 |
| | 200 | 0:54 | 28.74 | 0.869 |

Table 10: Inference times and corresponding performance for performing hard data consistency on with varying skip step sizes for chest CT reconstruction.

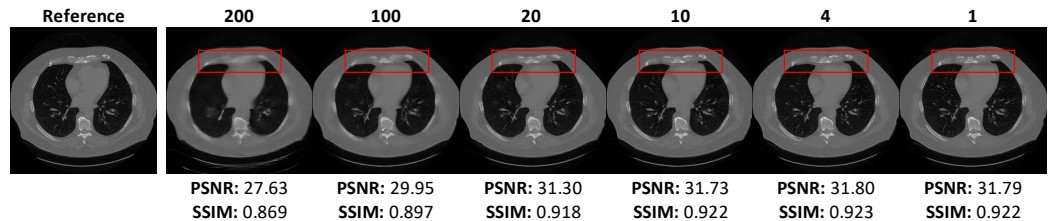

Figure 18: Observing the effect of the skip step size of hard data consistency optimization on CT reconstruction. The numbers above the image correspond to the number of hard data consistency steps per iteration of the reverse sampling process (e.g., 1 refers to hard data consistency on every step), and the red boxes outline the differences in reconstructions.

**Computational Efficiency.** Here, we present the memory usage and inference times of ReSample compared to our baselines in Tables 11 and 12. We observe that the memory gap between ReSample and other algorithms widens when the pre-trained diffusion model increases in size, as the hard data consistency step of ReSample requires minimal memory. Conversely, since we only apply hard data consistency in a subset of all sampling steps (which differs from existing methods that perform consistency updates at every time step), we note a slight increase in inference time compared to algorithms such as DPS, DMPS, and MCG. ReSample outperforms PSLD in terms of inference time.

Table 11: Average inference of different algorithms for Gaussian Deblurring

| Algorithm | Inference Time |
|---|---|
| DPS | 1:12 |
| MCG | 1:20 |
| DMPS | 1:25 |
| DDRM | 10 seconds |
| PSLD | 2:11 |
| ReSample (Ours) | 1:54 |

**Discussion on Latent-DPS.** In the main text, we briefly discussed how adding the Latent-DPS gradient term into our ReSample algorithm *can* improve the overall reconstruction quality. Generally, we observe that adding the Latent-DPS gradient will cause a very marginal boost in the PSNR, but only when the learning rate scale is chosen "correctly". Interestingly, we observe that choosing $\bar{\alpha}_t$ is *critical* in the performance of Latent-DPS. To this end, we perform a

Table 12: Memory Usage of different algorithms with different pretrained models for Gaussian Deblurring on FFHQ256 and ImageNet512 datasets

| Pretrained Model | Algorithm | Model Only | Memory Increment | Total Memory |
|---|---|---|---|---|
| DDPM(FFHQ) | DPS | **1953MB** | +3416MB (175%) | 5369MB |
| | MCG | | +3421MB (175%) | 5374MB |
| | DMPS | | +5215MB (267 %) | 7168MB |
| | DDRM | | +18833MB (964 %) | 20786MB |
| LDM(FFHQ) | PSLD | 3969MB | +5516MB (140%) | 9485MB |
| | ReSample (Ours) | | **+1040MB (26.2%)** | **5009MB** |
| DDPM(ImageNet) | DPS | **4394MB** | +6637MB (151%) | 11031MB |
| | MCG | | +6637MB (151%) | 11031MB |
| | DMPS | | +8731MB (199 %) | 13125MB |
| | DDRM | | +4530MB (103 %) | 8924MB |
| LDM(ImageNet) | PSLD | 5669MB | +5943MB (105%) | 11612MB |
| | ReSample (Ours) | | **+1322MB (30.1%)** | **7002MB** |

brief ablation study on the learning rate for Latent-DPS, where we choose the learning rate to be $k\bar{\alpha}_t$ for some $k > 0$. We vary $k$ and test the performance on 50 images of chest CT images and display the results in Table 13. In Table 13, we observe that $k = 0.5$ returns the best results, and should be chosen if one were to adopt the method of adding Latent-DPS into ReSample.

| $k$ | 0 | 0.5 | 1.0 | 1.5 | 2.0 | 2.5 |
|---|---|---|---|---|---|---|
| PSNR | 31.64 | **31.73** | 31.58 | 31.51 | 31.27 | 31.04 |
| Measurement Loss | **$9.82e-5$** | $1.21e-4$ | $5.03e-4$ | $1.35e-3$ | $1.96e-3$ | $2.63e-3$ |

Table 13: The effect of the Latent-DPS gradient scale (learning rate) as a function of $k > 0$. We study the PSNR and measurement loss (objective function) changes for varying $k$ for chest CT reconstruction on 50 test samples. The best results are in bold. Note that $k = 0$ here refers to no Latent-DPS and only using ReSample.

**Training Efficiency of LDMs.** To underscore the significance of our work, we conducted an ablation study focusing on the training efficiency of LDMs. This study demonstrates that LDMs require significantly fewer computational resources for training, a trait particularly beneficial for downstream applications like medical imaging. To support this assertion, we present the training time and performance of LDMs, comparing them to DDPMs (Ho et al., 2020) in the context of medical image synthesis. For LDMs, we utilized the pretrained autoencoder and LDM architecture provided by Rombach et al. (2022), training them on 9000 2D CT slices from three organs across 40 patients sourced from the LDCT training set Moen et al. (2021). In the case of DDPMs, we employed the codebase provided by Nichol & Dhariwal (2021) to train pixel-based diffusion models on the same set of 2D CT training slices. The results are presented in comparison with 300 2D slices from 10 patients in the validation set, detailed in Table 14. Observing the results in Table 14, we note that by utilizing LDMs, we can significantly reduce training time and memory usage while achieving better generation quality, as measured by the FID score.

| Model type | Training Time | Training Memory | FID |
|---|---|---|---|
| LDM (Ours) | **133 mins** | **7772MB** | **100.48** |
| DDPM | 229 mins | 10417MB | 119.50 |

Table 14: Comparison of training efficiency and performance of LDMs and DDPMs on CT images, computed on a V100 GPU.

# B    DISCUSSION ON HARD DATA CONSISTENCY

Recall that the hard data consistency step involves solving the following optimization problem:

$$\hat{z}_0(y) \in \arg\min_{z} \frac{1}{2}\|y - \mathcal{A}(\mathcal{D}(z))\|_2^2, \tag{14}$$

where we initialize using $\hat{z}_0(z_t)$. Instead of solving for the latent variable $z$ directly, notice that one possible technique would be to instead solve for vanilla least squares:

$$\hat{x}_0(y) \in \arg\min_{x} \frac{1}{2}\|y - \mathcal{A}(x)\|_2^2, \tag{15}$$

where we instead initialize using $\mathcal{D}(\hat{z}_0(z_t))$, where $\mathcal{D}(\cdot)$ denotes the decoder. Similarly, our goal here is to find the $\hat{x}_0(y)$ that is close to $\mathcal{D}(\hat{z}_0(z_t))$ that satisfies the measurement consistency: $\|y - \mathcal{A}(x)\|_2^2 < \sigma^2$, where $\sigma^2$ is an estimated noise level. Throughout the rest of the Appendix, we refer to the former optimization process as *latent optimization* and the latter as *pixel optimization*.

In our experiments, we actually found that these two different formulations yield different results, in the sense that performing latent optimization gives reconstructions that are "noisy", yet much sharper with fine details, whereas pixel optimization gives results that are "smoother" yet blurry with high-level semantic information. Here, the intuition is that pixel optimization does not directly change the latent variable where as latent optimization directly optimizes over the latent variable.

Moreover, the encoder $\mathcal{E}(\cdot)$ can add additional errors to the estimated $\hat{z}_0(\boldsymbol{y})$, perhaps throwing the sample off the data manifold, yielding images that are blurry as a result.

There is also a significant difference in time complexity between these two methods. Since latent optimization needs to backpropagate through the whole network of the decoder $\mathcal{D}(\cdot)$ on every gradient step, it takes much longer to obtain a local minimum (or converge). Empirically, to balance the trade-off between reconstruction speed and image quality, we see that using *both* of these formulations for hard data consistency can not only yield the best results, but also speed up the optimization process. Since pixel optimization is easy to get a global optimum, we use it first during the reverse sampling process and then use latent optimization when we are closer to $t = 0$ to refine the images with the finer details.

Lastly, we would like to remark that for pixel optimization, there is a closed-form solution that could be leveraged under specific settings Wang et al. (2022). If the forward operator $\mathcal{A}$ is linear and can take the matrix form $\mathbf{A}$ and the measurements are noiseless (i.e., $\boldsymbol{y} = \mathcal{A}(\hat{\boldsymbol{x}}_0(\boldsymbol{y}))$), then we can pose the optimization problem as

$$\hat{\boldsymbol{x}}_0(\boldsymbol{y}) \in \arg\min_{\boldsymbol{x}} \frac{1}{2}\|\mathcal{D}(\hat{\boldsymbol{z}}_0(\boldsymbol{z}_t)) - \boldsymbol{x}\|_2^2, \quad \text{s.t.} \quad \mathbf{A}\boldsymbol{x} = \boldsymbol{y}. \tag{16}$$

Then, the solution to this optimization problem is given by

$$\hat{\boldsymbol{x}}_0(\boldsymbol{y}) = \mathcal{D}(\hat{\boldsymbol{z}}_0(\boldsymbol{z}_t)) - (\mathbf{A}^+\mathbf{A}\mathcal{D}(\hat{\boldsymbol{z}}_0(\boldsymbol{z}_t)) - \mathbf{A}^+\boldsymbol{y}), \tag{17}$$

where by employing the encoder, we obtain

$$\hat{\boldsymbol{z}}_0(\boldsymbol{y}) = \mathcal{E}(\hat{\boldsymbol{x}}_0(\boldsymbol{y})) = \mathcal{E}(\mathcal{D}(\hat{\boldsymbol{z}}_0(\boldsymbol{z}_t)) - (\mathbf{A}^+\mathbf{A}\mathcal{D}(\hat{\boldsymbol{z}}_0(\boldsymbol{z}_t)) - \mathbf{A}^+\boldsymbol{y})). \tag{18}$$

This optimization technique does not require iterative solvers and offers great computational efficiency. In the following section, we provide ways in which we can compute a closed-form solution in the case in which the measurements may be noisy.

### B.1 ACCELERATED PIXEL OPTIMIZATION BY CONJUGATE GRADIENT

Notice that since pixel optimization directly operates in the pixel space, we can use solvers such as conjugate gradients least squares for linear inverse problems. Let $\mathbf{A}$ be the matrix form of the linear operator $\mathcal{A}(\cdot)$. Then, as discussed in the previous subsection, the solution to the optimization problem in the noiseless setting is given by

$$\hat{\boldsymbol{x}} = \boldsymbol{x}_0 - (\mathbf{A}^+\mathbf{A}\boldsymbol{x}_0 - \mathbf{A}^+\boldsymbol{y}), \tag{19}$$

where $\mathbf{A}^+ = \mathbf{A}^\top(\mathbf{A}\mathbf{A}^\top)^{-1}$ and $(\mathbf{A}\mathbf{A}^\top)^{-1}$ can be implemented by conjugate gradients. In the presence of measurement noise, we can relax this solution to

$$\hat{\boldsymbol{x}} = \boldsymbol{x}_0 - \kappa(\mathbf{A}^+\mathbf{A}\boldsymbol{x}_0 - \mathbf{A}^+\boldsymbol{y}), \tag{20}$$

$\kappa \in (0, 1)$ is a hyperparameter that can reduce the impact between the noisy component of the measurements and $\boldsymbol{x}_0$ (the initial image before optimization, for which we use $\mathcal{D}(\hat{\boldsymbol{z}}_0(\boldsymbol{z}_t))$).

We use this technique for CT reconstruction, where the forward operator $\mathbf{A}$ is the radon transform and $\mathbf{A}^\top$ is the non-filtered back projection. However, we can use this conjugate gradient technique for any linear inverse problem where the matrix $\mathbf{A}$ is available.

## C IMPLEMENTATION DETAILS

In this section, we discuss the choices of the hyperparameters used in all of our experiments for our algorithm. All experiments are implemented in `PyTorch` on NVIDIA GPUs (A100 and A40).

### C.1 LINEAR AND NONLINEAR INVERSE PROBLEMS ON NATURAL IMAGES

For organizational purposes, we tabulate all of the hyperparameters associated to ReSample in Table 15. The parameter for the number of times to perform hard data consistency is not included in the table, as we did not have any explicit notation for it. For experiments across all natural images

| Notation | Definition |
|---|---|
| $\tau$ | Hyperparameter for early stopping in hard data consistency |
| $\sigma_t$ | Variance scheduling for the resampling technique |
| $T$ | Number of DDIM or DDPM steps |

Table 15: Summary of the hyperparameters with their respective notations for `ReSample`.

datasets (LSUN-Bedroom, FFHQ, CelebA-HQ), we used the same hyperparameters as they seemed to all empirically give the best results.

For $T$, we used $T = 500$ DDIM steps. For hard data consistency, we first split $T$ into three even sub-intervals. The first stage refers to the sub-interval closest to $t = T$ and the third stage refers to the interval closest to $t = 0$. During the second stage, we performed pixel optimization, whereas in the third stage we performed latent optimization for hard data consistency as described in Section B. We performed this optimization on every 10 iterations of $t$.

We set $\tau = 10^{-4}$, which seemed to give us the best results for noisy inverse problems, with a maximum number of iterations of 2000 for pixel optimization and 500 for latent optimization (whichever convergence criteria came first). For the variance hyperparameter $\sigma_t$ in the stochastic resample step, we chose an adaptive schedule of

$$\sigma_t^2 = \gamma \left( \frac{1 - \bar{\alpha}_{t-1}}{\bar{\alpha}_t} \right) \left( 1 - \frac{\bar{\alpha}_t}{\bar{\alpha}_{t-1}} \right),$$

as discussed in Section A. Generally, we see that $\gamma = 40$ returns the best results for experiments on natural images.

## C.2 LINEAR INVERSE PROBLEMS ON MEDICAL IMAGES

Since the LDMs for medical images is not as readily available as compared to natural images, we largely had to fine-tune existing models. We discuss these in more detail in this section.

**Backbone Models.** For the backbone latent diffusion model, we use the pre-trained model from latent diffusion (Rombach et al., 2022). We select the VQ-4 autoencoder and the FFHQ-LDM with CelebA-LDM as our backbone LDM. For inferencing, upon taking pre-trained checkpoints provided by Rombach et al. (2022), we fine-tuned the models on 2000 CT images with 100K iterations and a learning rate of $10^{-5}$.

**Inferencing.** For $T$, we used a total of $T = 1000$ DDIM steps. For hard data consistency, we split $T$ into three sub-intervals: $t > 750$, $300 < t \leq 750$, and $t \leq 300$. During the second stage, we performed pixel optimization by using conjugate gradients as discussed previously, with 50 iterations with $\kappa = 0.9$. In the third stage, we performed latent optimization with $\tau$ as the estimated noise level $\tau = 10^{-4}$. We set skip step size to be 10 and $\gamma = 40$, with $\sigma_t$ as the same as the experiments for the natural images.

## C.3 IMPLEMENTATIONS OF BASELINES

**Latent-DPS.** For the Latent-DPS baseline, we use $T = 1000$ DDIM steps for CT reconstruction and $T = 500$ DDIM steps for natural image experiments. Let $\zeta_t$ denote the learning rate. For medical images, we use $\zeta_t = 2.5\bar{\alpha}_t$ and $\zeta_t = 0.5\bar{\alpha}_t$ for natural images. Empirically, we observe that our proposed $\zeta_t$ step size schedule gives the best performance and is robust to scale change, as previously discussed.

**DPS and MCG.** For DPS, we use the original DPS codebase provided by Chung et al. (2023a) and pre-trained models trained on CelebA and FFHQ training sets for natural images. For medical images, we use the pretrained checkpoint from Chung et al. (2022) on the dataset provided by Moen et al. (2021). For MCG (Chung et al., 2022), we modified the MCG codebase by deleting the projection term tuning the gradient term for running DPS experiments on CT reconstruction. Otherwise, we directly used the codes provided by Chung et al. (2022) for both natural and medical image experiments.

**DDRM.** For DDRM, we follow the original code provided by Kawar et al. (2022) withDDPM models trained on FFHQ and CelebA training sets adopted from the repository provided by Dhariwal & Nichol (2021b). We use the default parameters as displayed by Kawar et al. (2022).

**DMPS.** We follow the original code from the repository of Meng & Kabashima (2022) with the DDPM models trained on FFHQ and CelebA training sets adopted from the repository of Dhariwal & Nichol (2021b). We use the default parameters as displaye d by Meng & Kabashima (2022).

**PSLD.** We follow the original code from the repo Shah et al. (2021) with the pretrained LDMs on CelebA and FFHQ datasets provided by Rombach et al. (2022). We use the default hyperparameters as implied in Shah et al. (2021).

**ADMM-PnP and Other (Supervised) Baselines.** For ADMM-PnP we use the 10 iterations with $\tau$ tuned for different inverse problems. We use $\tau = 5$ for CT reconstruction, $\tau = 0.1$ for linear inverse problems, and $\tau = 0.075$ for nonlinear deblurring. We use the pre-trained model from original DnCNN repository provided by Zhang et al. (2017). We observe that ADMM-PnP tends to over-smooth the images with more iterations, which causes performance degradation. For FBP-UNet, we trained a UNet that maps FBP images to ground truth images. The UNet network architecture is the same as the one explained by Jin et al. (2017).

## D    MORE DISCUSSION ON THE FAILURE CASES OF LATENT-DPS

In this section, we provide a further explanation to which why Latent-DPS often fails to give accurate reconstructions.

### D.1    FAILURE OF MEASUREMENT CONSISTENCY

Previously, we claimed that by using Latent-DPS, it is likely that we converge to a local minimum of the function $\|\mathbf{y} - \mathcal{A}(\mathcal{D}(\hat{z}_0))\|_2^2$ and hence cannot achieve accurate measurement consistency. Here, we validate this claim by comparing the measurement consistency loss between Latent-DPS and ReSample. We observe that ReSample is able to achieve better measurement consistency than Latent-DPS. This observation validates our motivation of using hard data consistency to improve reconstruction quality.

| Anatomical Site | ReSample (Ours) | Latent-DPS |
|:---:|:---:|:---:|
| Chest | **5.36e-5** | 2.99e-3 |
| Abdominal | **6.28e-5** | 3.92e-3 |
| Head | **1.73e-5** | 2.03e-3 |

Table 16: Comparison of average measurement consistency loss for CT reconstruction between ReSample and Latent-DPS

### D.2    FAILURE OF LINEAR MANIFOLD ASSUMPTION

We hypothesize that one reason that Latent-DPS fails to give accurate reconstructions could due to the nonlinearity of the decoder $\mathcal{D}(\cdot)$. More specifically, the derivation of DPS provided by Chung et al. (2023a) relies on a *linear manifold assumption*. For our case, since our forward model can be viewed as the form $\mathcal{A}(\mathcal{D}(\cdot))$, where the decoder $\mathcal{D}(\cdot)$ is a highly nonlinear neural network, this linear manifold assumption fails to hold. For example, if two images $z^{(1)}$ and $z^{(2)}$ lie on the clean data distribution manifold $\mathcal{M}$ at $t = 0$, a linear combination of them $az^{(1)} + bz^{(2)}$ for some constants $a$ and $b$, may not belong to $\mathcal{M}$ since $\mathcal{D}(az^{(1)} + bz^{(2)})$ may not give us an image that is realistic. Thus, in practice, we observe that DPS reconstructions tend to be more blurry, which implies that the reverse sampling path falls out of this data manifold. We demonstrate this in Figure 19, where we show that the average of two latent vectors gives a blurry and unrealistic image.

We would like to point out that this reasoning may also explain why our algorithm outperforms DPS on nonlinear inverse tasks.

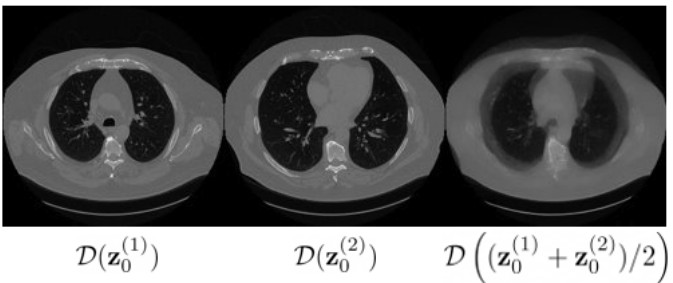

$$\mathcal{D}(\mathbf{z}_0^{(1)}) \qquad \mathcal{D}(\mathbf{z}_0^{(2)}) \qquad \mathcal{D}\left((\mathbf{z}_0^{(1)} + \mathbf{z}_0^{(2)})/2\right)$$

Figure 19: Depiction of the violation of the linear manifold assumption.

### D.3    INACCURATE ESTIMATES OF THE POSTERIOR MEAN

In this section, we discuss how inaccurate estimates of $\hat{z}_0(z_t)$ (i.e., the posterior mean via Tweedie's formula) can be one of the reasons why Latent-DPS returns image reconstructions that are noisy. This was mainly because for values of $t$ closer to $t = T$, the estimate of $\hat{z}_0(z_t)$ may be inaccurate, leading us to images that are noisy at $t = 0$. Generally, we find that the estimation of $\hat{z}_0(z_t)$ is

inaccurate in the early timesteps (e.g. $t > 0.5T$) and vary a lot when $t$ decreases. This implies that the gradient update may not point to a consistent direction when $t$ is large. We further demonstrate this observation in Figure 20.

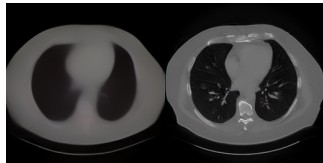

Figure 20: Comparison of the prediction of the ground truth signal $\hat{z}_0(z_t)$ for different values of $t$. Left: $\hat{z}_0(z_t)$ when $t = 0.5T$. Right: $\hat{z}_0(z_t)$ when $t = 0$. This serves to show the estimation error of the posterior mean for large values of $t$ (i.e. when $t$ is closer to pure noise).

## E    RELATED WORKS

Deep neural networks have been extensively employed as priors for solving inverse problems (Han & Ye, 2018; Bora et al., 2017; Zhu et al., 2018; Gupta et al., 2018). Numerous works focus on learning the mapping between measurements and clean images, which we term as **supervised** methods (Han & Ye, 2018; Zhu et al., 2018; Wei et al., 2020; Liang et al., 2021). Supervised methods necessitate training on pairs of measurements and clean images, requiring model retraining for each new task. On the other hand, another line of research aims at learning the prior distribution of ground truth images, solving inverse problems at inference time without retraining by using the pre-trained prior distributions (Bora et al., 2017; Jalal et al., 2021; Hussein et al., 2020; Lempitsky et al., 2018). We categorize this as **unsupervised** methods.

Until the advent of diffusion models, unsupervised methods struggled to achieve satisfactory reconstruction quality compared to supervised methods (Jalal et al., 2021). These methods rely on optimization within a constrained space (Bora et al., 2017), while constructing a space that accurately encodes the ground truth data distribution to solve the optimization problem is challenging. However, with the accurate approximation of the prior distribution now provided by diffusion models, unsupervised methods can outperform supervised methods for solving inverse problems more efficiently without retraining the model (Song et al., 2021; Jalal et al., 2021).

For unsupervised approaches using diffusion models as priors, the plug-and-play approach has been widely applied for solving linear inverse problems (Song et al., 2021; Kawar et al., 2022; Wang et al., 2022). These methods inject the measurement on the noisy manifold or incorporate an estimate of the ground truth image into the reverse sampling procedure. We refer to these methods as **hard** data consistency approaches, as they directly inject the measurements into the reverse sampling process. However, to the best of our knowledge, only a few of these methods extend to nonlinear inverse problems or even latent diffusion models (Rombach et al., 2022). Other approaches focus on approximating the conditional score (Dhariwal & Nichol, 2021b; Chung et al., 2023a; 2022) under some mild assumptions using gradient methods. While these methods can achieve excellent performance and be extended to nonlinear inverse problems, measurement consistency may be compromised, as demonstrated in our paper. Since these methods apply data consistency only through a gradient in the reverse sampling process, we refer to these methods as **soft** data consistency approaches.

Recently, there has been a growing interest in modeling the data distribution by diffusion models in a latent space or a constraint domain Liu et al. (2023); Rombach et al. (2022); Lou & Ermon (2023b); Fishman et al. (2023). There has also been a growing interest in solving inverse problems using latent diffusion models. In particular, Rout et al. (2023) proposed PSLD, an unsupervised soft approach that adds a gradient term at each reverse sampling step to solve linear inverse problems with LDMs. However, there are works that observe that this approach may suffer from instability due to estimating the gradient term Chung et al. (2023c). Our work proposes an unsupervised hard approach that involves "resampling" and enforcing hard data consistency for solving general inverse problems (both linear and nonlinear) using latent diffusion models. By using a hard data consistency approach, we can obtain much better reconstructions, highlighting the effectiveness of our algorithm.

# F    DEFERRED PROOFS FOR RESAMPLE

Here, we provide the proofs regarding the theory behind our resampling technique. To make this section self-contained, we first restate all of our results and discuss notation that will be used throughout this section.

**Notation.** We denote scalars with under-case letters (e.g. $\alpha$) and vectors with bold under-case letters (e.g. $\boldsymbol{x}$). Recall that in the main body of the paper, $\boldsymbol{z} \in \mathbb{R}^k$ denotes a sample in the latent space, $\boldsymbol{z}'_t$ denotes an unconditional sample at time step $t$, $\hat{\boldsymbol{z}}_0(\boldsymbol{z}_t)$ denotes a prediction of the ground truth signal $\boldsymbol{z}_0$ at time step $t$, $\hat{\boldsymbol{z}}_0(\boldsymbol{y})$ denotes the measurement-consistent sample of $\hat{\boldsymbol{z}}_0(\boldsymbol{z}_t)$ using hard data consistency, and $\hat{\boldsymbol{z}}_t$ denotes the re-mapped sample from $\hat{\boldsymbol{z}}_0(\boldsymbol{y})$ onto the data manifold at time step $t$. We use $\hat{\boldsymbol{z}}_t$ as the next sample to resume the reverse diffusion process.

**_Proposition_** 1 (Stochastic Encoding). Since the sample $\hat{\boldsymbol{z}}_t$ given $\hat{\boldsymbol{z}}_0(\boldsymbol{y})$ and measurement $\boldsymbol{y}$ is conditionally independent of $\boldsymbol{y}$, we have that

$$p(\hat{\boldsymbol{z}}_t|\hat{\boldsymbol{z}}_0(\boldsymbol{y}), \boldsymbol{y}) = p(\hat{\boldsymbol{z}}_t|\hat{\boldsymbol{z}}_0(\boldsymbol{y})) = \mathcal{N}(\sqrt{\bar{\alpha}_t}\hat{\boldsymbol{z}}_0(\boldsymbol{y}), (1 - \bar{\alpha}_t)\boldsymbol{I}). \tag{21}$$

*Proof.* By Tweedie's formula, we have that $\hat{\boldsymbol{z}}_0(\boldsymbol{y})$ is the estimated mean of the ground truth signal $\boldsymbol{z}_0$. By the forward process of the DDPM formulation Ho et al. (2020), we also have that

$$p(\hat{\boldsymbol{z}}_t|\hat{\boldsymbol{z}}_0(\boldsymbol{y})) = \mathcal{N}(\sqrt{\bar{\alpha}_t}\hat{\boldsymbol{z}}_0(\boldsymbol{y}), (1 - \bar{\alpha}_t)\boldsymbol{I}).$$

Then, since $\boldsymbol{y}$ is a measurement of $\boldsymbol{z}_0$ at $t = 0$, we have $p(\boldsymbol{y}|\hat{\boldsymbol{z}}_0(\boldsymbol{y}), \hat{\boldsymbol{z}}_t) = p(\boldsymbol{y}|\hat{\boldsymbol{z}}_0(\boldsymbol{y}))$. Finally, we get

$$p(\hat{\boldsymbol{z}}_t|\hat{\boldsymbol{z}}_0(\boldsymbol{y}), \boldsymbol{y}) = \frac{p(\boldsymbol{y}|\hat{\boldsymbol{z}}_t, \hat{\boldsymbol{z}}_0(\boldsymbol{y}))p(\hat{\boldsymbol{z}}_t|\hat{\boldsymbol{z}}_0(\boldsymbol{y}))p(\hat{\boldsymbol{z}}_0(\boldsymbol{y}))}{p(\boldsymbol{y}, \hat{\boldsymbol{z}}_0(\boldsymbol{y}))} \tag{22}$$

$$= \frac{p(\boldsymbol{y}, \hat{\boldsymbol{z}}_0(\boldsymbol{y}))p(\hat{\boldsymbol{z}}_t|\hat{\boldsymbol{z}}_0(\boldsymbol{y}))}{p(\boldsymbol{y}, \hat{\boldsymbol{z}}_0(\boldsymbol{y}))} \tag{23}$$

$$= p(\hat{\boldsymbol{z}}_t|\hat{\boldsymbol{z}}_0(\boldsymbol{y})). \tag{24}$$

$\square$

**_Proposition_** 2 (Stochastic Resampling). Suppose that $p(\boldsymbol{z}'_t|\hat{\boldsymbol{z}}_t, \hat{\boldsymbol{z}}_0(\boldsymbol{y}), \boldsymbol{y})$ is normally distributed such that $p(\boldsymbol{z}'_t|\hat{\boldsymbol{z}}_t, \hat{\boldsymbol{z}}_0(\boldsymbol{y}), \boldsymbol{y}) = \mathcal{N}(\mu_t, \sigma_t^2)$. If we let $p(\hat{\boldsymbol{z}}_t|\hat{\boldsymbol{z}}_0(\boldsymbol{y}), \boldsymbol{y})$ be a prior for $\boldsymbol{\mu}_t$, then the posterior distribution $p(\hat{\boldsymbol{z}}_t|\boldsymbol{z}'_t, \hat{\boldsymbol{z}}_0(\boldsymbol{y}), \boldsymbol{y})$ is given by

$$p(\hat{\boldsymbol{z}}_t|\boldsymbol{z}'_t, \hat{\boldsymbol{z}}_0(\boldsymbol{y}), \boldsymbol{y}) = \mathcal{N}\left(\frac{\sigma_t^2\sqrt{\bar{\alpha}_t}\hat{\boldsymbol{z}}_0(\boldsymbol{y}) + (1 - \bar{\alpha}_t)\boldsymbol{z}'_t}{\sigma_t^2 + (1 - \bar{\alpha}_t)}, \frac{\sigma_t^2(1 - \bar{\alpha}_t)}{\sigma_t^2 + (1 - \bar{\alpha}_t)}\boldsymbol{I}\right). \tag{25}$$

*Proof.* We have that

$$p(\hat{\boldsymbol{z}}_t|\boldsymbol{z}'_t, \hat{\boldsymbol{z}}_0, \boldsymbol{y}) = \frac{p(\boldsymbol{z}'_t|\hat{\boldsymbol{z}}_t, \hat{\boldsymbol{z}}_0, \boldsymbol{y})p(\hat{\boldsymbol{z}}_t|\hat{\boldsymbol{z}}_0, \boldsymbol{y})p(\hat{\boldsymbol{z}}_0, \boldsymbol{y})}{p(\boldsymbol{z}'_t, \hat{\boldsymbol{z}}_0, \boldsymbol{y})}, \tag{26}$$

where both $p(\boldsymbol{z}'_t, \hat{\boldsymbol{z}}_0, \boldsymbol{y})$ and $(\hat{\boldsymbol{z}}_0, \boldsymbol{y})$ are normalizing constants. Then by Lemma 1, we have $p(\hat{\boldsymbol{z}}_t|\hat{\boldsymbol{z}}_0, \boldsymbol{y}) = \mathcal{N}(\sqrt{\bar{\alpha}_t}\hat{\boldsymbol{z}}_0, (1 - \bar{\alpha}_t)\boldsymbol{I})$. Now, we can compute the posterior distribution:

$$p(\hat{\boldsymbol{z}}_t = \boldsymbol{k}|\boldsymbol{z}'_t, \hat{\boldsymbol{z}}_0, \boldsymbol{y}) \propto p(\boldsymbol{z}'_t|\hat{\boldsymbol{z}}_t = \boldsymbol{k}, \hat{\boldsymbol{z}}_0, \boldsymbol{y})p(\hat{\boldsymbol{z}}_t = \boldsymbol{k}|\hat{\boldsymbol{z}}_0, \boldsymbol{y}) \tag{27}$$

$$\propto \exp\left\{-\left(\frac{(\boldsymbol{k} - \boldsymbol{z}'_t)^2}{\sigma_t^2}\right) + \left(\frac{(\boldsymbol{k} - \sqrt{\bar{\alpha}_t}\hat{\boldsymbol{z}}_0)^2}{1 - \bar{\alpha}_t}\right)\right\} \tag{28}$$

$$\propto \exp\left\{\frac{-\left(\boldsymbol{k} - \frac{(1-\bar{\alpha}_t)\boldsymbol{z}'_t + \sqrt{\bar{\alpha}_t}\hat{\boldsymbol{z}}_0\sigma_t^2}{\sigma_t^2 + (1-\bar{\alpha}_t)}\right)^2}{\frac{1}{\sigma_t^2} + \frac{1}{1-\bar{\alpha}_t}}\right\}. \tag{29}$$

This as a Gaussian distribution, which can be easily shown using moment-generating functions Bromiley (2013). $\square$

**Theorem** 1. If $\hat{z}_0(y)$ is measurement-consistent such that $y = \mathcal{A}(\mathcal{D}(\hat{z}_0(y)))$, i.e. $\hat{z}_0 = \hat{z}_0(z_{t+1}) = \hat{z}_0(y)$, then stochastic resample is unbiased such that $\mathbb{E}[\hat{z}_t|y] = \mathbb{E}[z_t']$.

*Proof.* We have that $\mathbb{E}[\hat{z}_t|y] = \mathbb{E}_{z_t'}[\mathbb{E}_{\hat{z}_0(y)}[\mathbb{E}_{\hat{z}_t}[\hat{z}_t|z_t', \hat{z}_0(y), y]]]$.

Let $\gamma = \frac{\sigma_t^2}{\sigma_t^2 + 1 - \bar{\alpha}_t}$. By using Proposition 2, we have $\mathbb{E}[\hat{z}_t|y] = \mathbb{E}_{z_t'}[\mathbb{E}_{\hat{z}_0}[(\gamma\sqrt{\bar{\alpha}_t}\hat{z}_0 + (1 - \gamma)z_t')|\hat{z}_0, z_t', y]]$.

Since $\hat{z}_0$ is measurement-consistent such that $y = \mathcal{A}(\mathcal{D}(\hat{z}_0(y)))$, let $k = -1$, we have

$$\hat{z}_0(y) = \hat{z}_0 = \frac{1}{\sqrt{\bar{\alpha}_{t-k}}}(z_{t-k}' + (1 - \bar{\alpha}_{t-k})\nabla \log p(z_{t-k}')) \tag{30}$$

Then, we have that

$$\mathbb{E}[\hat{z}_t|y] = \mathbb{E}_{z_t'}[\mathbb{E}_{\hat{z}_0}[(\gamma\sqrt{\bar{\alpha}_t}\hat{z}_0 + (1 - \gamma)z_t')|\hat{z}_0, z_t', y]] \tag{31}$$

$$= \gamma\sqrt{\frac{\bar{\alpha}_t}{\bar{\alpha}_{t-k}}}\mathbb{E}_{z_t'}[\mathbb{E}[z_{t-k}' + (1 - \bar{\alpha}_{t-k})\nabla \log p(z_{t-k}')|z_t']] + (1 - \gamma)\mathbb{E}[z_t'], \tag{32}$$

as both $z_t'$ and $z_{t-k}'$ are unconditional samples and independent of $y$. Now, we have

$$\mathbb{E}_{z_t'}[\mathbb{E}[z_{t-k}' + (1 - \bar{\alpha}_{t-k})\nabla \log p(z_{t-k}')|z_t']] = \mathbb{E}_{z_{t-k}'}[z_{t-k}' + (1 - \bar{\alpha}_{t-k})\nabla \log p(z_{t-k}')] \tag{33}$$

Since $z_t'$ is the unconditional reverse sample of $z_{t-k}'$, we have $\mathbb{E}[z_t'] = \sqrt{\frac{\bar{\alpha}_t}{\bar{\alpha}_{t-k}}}\mathbb{E}[z_{t-k}']$, and then

$$\mathbb{E}_{z_{t-k}'}[\nabla \log p(z_{t-k}')] = \int \nabla \log p(z_{t-k}')p(z_{t-k}')dz_{t-k}' \tag{34}$$

$$= \int \frac{p'(z_{t-k}')}{p(z_{t-k}')}p(z_{t-k}')dz_{t-k}' \tag{35}$$

$$= \frac{\partial(1)}{\partial z_{t-k}'} = 0 \tag{36}$$

Finally, we have $\mathbb{E}[\hat{z}_t|y] = \gamma\mathbb{E}[z_t'] + (1 - \gamma)\mathbb{E}[z_t'] = \mathbb{E}[z_t']$. $\square$

**Lemma** 1. Let $\tilde{z}_t$ and $\hat{z}_t$ denote the stochastically encoded and resampled image of $\hat{z}_0(y)$, respectively. If $\text{VAR}(z_t') > 0$, then we have that $\text{VAR}(\hat{z}_t) < \text{VAR}(\tilde{z}_t)$.

*Proof.* Recall that $\hat{z}_t$ and $\tilde{z}_t$ are both normally distributed, with

$$\text{VAR}(\hat{z}_t) = 1 - \bar{\alpha}_t \tag{37}$$

$$\text{VAR}(\tilde{z}_t) = \frac{1}{\frac{1}{1-\bar{\alpha}_t} + \frac{1}{\sigma_t^2}} \tag{38}$$

$$= \frac{\sigma_t^2(1 - \bar{\alpha}_t)}{\sigma_t^2 + (1 - \bar{\alpha}_t)}. \tag{39}$$

For all $\sigma_t^2 \geq 0$, we have

$$\frac{\sigma_t^2(1 - \bar{\alpha}_t)}{\sigma_t^2 + (1 - \bar{\alpha}_t)} < 1 - \bar{\alpha}_t \tag{40}$$

$$\implies \text{VAR}(\hat{z}_t) < \text{VAR}(\tilde{z}_t). \tag{41}$$

$\square$

**Theorem** 2. Let $z_0$ denote a sample from the data distribution and $z_t$ be a sample from the noisy perturbed distribution at time $t$. Given that the score function $\nabla_{z_t} \log p_{z_t}(z_t)$ is bounded,

$$\text{Cov}(z_0|z_t) = \frac{(1 - \bar{\alpha}_t)^2}{\bar{\alpha}_t}\nabla_{z_t}^2 \log p_{z_t}(z_t) + \frac{1 - \bar{\alpha}_t}{\bar{\alpha}_t}I,$$

where $\alpha_t \in (0, 1)$ is an decreasing sequence in $t$.

*Proof.* By the LDM forward process, we have

$$p(\boldsymbol{z}_t|\boldsymbol{z}_0) = \mathcal{N}(\sqrt{\bar{\alpha}_t}, 1 - \bar{\alpha}_t).$$

Consider the following variable: $\hat{\boldsymbol{z}}_t = \frac{\sqrt{\bar{\alpha}_t}}{1-\bar{\alpha}_t}\boldsymbol{z}_t$, which is simply a scaled version of $\boldsymbol{z}_t$. Then, consider the distribution

$$p(\bar{\boldsymbol{z}}_t|\boldsymbol{z}_0) = \frac{1}{\left(2\pi\left(\frac{\bar{\alpha}_t}{1-\bar{\alpha}_t}\right)\right)^{d/2}} \exp\left\{-\frac{\|\bar{\boldsymbol{z}}_t - \frac{\bar{\alpha}_t}{1-\bar{\alpha}_t}\boldsymbol{z}_0\|_2^2}{2\left(\frac{\bar{\alpha}_t}{1-\bar{\alpha}_t}\right)}\right\}.$$

Now, we want to separate the term only with $\bar{\boldsymbol{z}}_t$ and the term only with $\boldsymbol{z}_0$ for applying Tweedie's formula. Hence, let $p_0(\bar{\boldsymbol{z}}_t) = \frac{1}{\left(2\pi\left(\frac{\bar{\alpha}_t}{1-\bar{\alpha}_t}\right)\right)^{d/2}} \exp\left(-\frac{\|\bar{\boldsymbol{z}}_t\|^2}{2\left(\frac{\bar{\alpha}_t}{1-\bar{\alpha}_t}\right)}\right)$, we achieve

$$p(\bar{\boldsymbol{z}}_t|\boldsymbol{z}_0) = p_0(\bar{\boldsymbol{z}}_t)\exp\left\{\bar{\boldsymbol{z}}_t^\top \boldsymbol{z}_0 - \frac{\bar{\alpha}_t}{2(1-\bar{\alpha}_t)}\|\boldsymbol{z}_0\|^2\right\},$$

which separates the distribution interaction term $\bar{\boldsymbol{z}}_t^\top \boldsymbol{z}_0$ from $\|\boldsymbol{z}_0\|^2$ term.
Let $\lambda(\bar{\boldsymbol{z}}_t) = \log p(\bar{\boldsymbol{z}}_t) - \log p_0(\bar{\boldsymbol{z}}_t)$. Then by Tweedie's formula, we have $\mathbb{E}[\boldsymbol{z}_0|\bar{\boldsymbol{z}}_t] = \nabla\lambda(\bar{\boldsymbol{z}}_t)$ and $\text{Cov}(\boldsymbol{z}_0|\bar{\boldsymbol{z}}_t) = \nabla^2\lambda(\bar{\boldsymbol{z}}_t)$. Since $p_0(\bar{\boldsymbol{z}}_t)$ is a Gaussian distribution with mean at $0$ and variance equal to $\frac{\bar{\alpha}_t}{1-\bar{\alpha}_t}$, we obtain

$$\nabla\lambda(\bar{\boldsymbol{z}}_t) = \nabla \log p(\bar{\boldsymbol{z}}_t) + \frac{1-\bar{\alpha}_t}{\bar{\alpha}_t}\bar{\boldsymbol{z}}_t.$$

We observe that since $\hat{\boldsymbol{z}}$ is a scaled version of $\boldsymbol{z}_t$, we can obtain the distribution of $\hat{\boldsymbol{z}}$ as

$$p(\bar{\boldsymbol{z}}_t) = \frac{1-\bar{\alpha}_t}{\sqrt{\bar{\alpha}_t}}p\left(\frac{1-\bar{\alpha}_t}{\sqrt{\bar{\alpha}_t}}\cdot\boldsymbol{z}_t\right).$$

Then, we can apply chain rule to first compute then gradient on $\boldsymbol{z}_t$, and then account for $\bar{\boldsymbol{z}}_t$. As a result we get

$$\nabla_{\bar{\boldsymbol{z}}_t} \log p(\bar{\boldsymbol{z}}_t) = \frac{1-\bar{\alpha}_t}{\sqrt{\bar{\alpha}_t}}\nabla_{\boldsymbol{z}_t} \log p(\boldsymbol{z}_t)$$

and then

$$\nabla\lambda(\bar{\boldsymbol{z}}_t) = \mathbb{E}[\boldsymbol{z}_0|\boldsymbol{z}_t] = \frac{1-\bar{\alpha}_t}{\sqrt{\bar{\alpha}_t}}\nabla \log p(\boldsymbol{z}_t) + \frac{1}{\sqrt{\bar{\alpha}_t}}\boldsymbol{z}_t,$$

which is consistent with the score function of Chung et al. (2023a). Afterwards, we take the gradient again with respective to the $\bar{\boldsymbol{z}}_t$, and apply the chain rule again, then we have

$$\nabla^2\lambda(\bar{\boldsymbol{z}}_t) = \text{Cov}(\boldsymbol{z}_0|\boldsymbol{z}_t) = \frac{(1-\bar{\alpha}_t)^2}{\bar{\alpha}_t}\nabla^2 \log p(\boldsymbol{z}_t) + \frac{1-\bar{\alpha}_t}{\bar{\alpha}_t}\boldsymbol{I}.$$

This gives the desired result. $\qquad\square$

