# OpenReview forum: "Solving Inverse Problems with Latent Diffusion Models via Hard Data Consistency"
_ICLR.cc/2024/Conference — ICLR 2024 spotlight_

### Official Review · Reviewer_4YEX · 2023-10-21

**Soundness:** 3 good
**Presentation:** 3 good
**Contribution:** 3 good
**Rating:** 8
**Confidence:** 3

**Summary:**

In this paper, the ReSample algorithm is introduced, leveraging the capabilities of diffusion models to address inverse problems. Unlike other approaches using diffusion models for inverse problems, ReSample trains the model in a lower-dimensional latent space, offering a computational advantage over conventional diffusion models trained in pixel space. Although a similar method was introduced very recently by Rout et al. in 2023, ReSample enhances its performance by introducing innovative sampling techniques supported by theoretical justification and validated through numerous experiments involving both linear and non-linear operators.

**Strengths:**

-	A design of a novel sampling method supported by theory. It is proven that the proposed sampling method is less noisy than stochastic sampling.

-	The paper is well organized.

**Weaknesses:**

-	The idea of applying a latent space to solve inverse problems using diffusion models is not groundbreaking. Nonetheless, the paper introduces significant advancements to this approach.

-	It is not clear why training the model in pixel space (Table 1 and 2) achieves lower quality results compared with the proposed ReSample algorithm. The rationale behinds this result remains unclear.

-	The experimental results primarily involve random inpainting, a task considered relatively straightforward. It would be valuable to explore how the proposed algorithm performs in more complex inpainting scenarios, such as the removal of large pixel regions, as demonstrated in the approach outlined by Rout et al. in 2023.

-	A minor issue to address: There is a typographical error in the term "noice" after Theorem 2.

**Questions:**

-	The primary incentive for training the model in a lower-dimensional latent space appears to be a reduction in computational complexity. Nevertheless, the experimental outcomes presented in Table 1 and 2 demonstrate that the proposed algorithm surpasses conventional techniques in terms of reconstruction quality. Could you provide insights into the potential factors contributing to this outcome?

---

> ### Author Response · Authors · 2023-11-16
> **Official Response from authors of paper 4764**
>
> We would like to thank the reviewer for the insightful comments! Below, we address the reviewer’s concerns in detail:
>
> ***Comment:*** the rationale behind the result: why pixel-space diffusion models achieves lower quality than with our method
>
>
>
> ***Response:***  Thanks for pointing this out! We have the follow hypotheses:
> 1. LDM has potential to achieve better generation quality than pixel-based diffusion models. This is highlighted in the paper [1] . It shows that LDM is able to outperform DDPM based methods in FID when training on the same dataset. This indicates that LDM may approximate the prior distribution better than pixel-based DDPM models.
> 2. The gain of our performance may comes from the careful design of resample and hard consistency, which may leads to smaller measurement consistency loss compared to DPS-based methods especially when the forward operator is nonlinear and nonconvex. As shown in Table I in the following, we observe that ReSample is able to achieve better measurement consistency than latentDPS. Also DPS shows significant performance degradation when it comes to nonlinear inverse problems, but the performance of ReSample is potentially more stable as demonstrated in Table 2 and 6 in the paper. This observation further corroborates our hypothesis. We provide insights on this in Appendix section D.
> 3. Some work in parallel with ours theoretically demonstrates that DPS-based algorithms may encounter performance degradation with higher resolution data, as reported in [2].
>
>
> |Anatomical site|ReSample (Ours)|Latent-DPS|
> | -------- | ------- |------- |
> |Chest	|***5.36e-5***	|2.99e-3|
> |Abdominal|	***6.28e-5***|	3.92e-3|
> |Head	|***1.73e-5***	|2.03e-3|
>
> Table I: average measurement consistency loss for CT reconstruction for ReSample and Latent-DPS
>
>
> ***Comment:*** Experimental Results on box-inpainting and comparison with PSLD
>
>
> ***Response:***  Thanks for the suggestion! We perform additional experiments for box inpainting task on the CelebA-HQ dataset. We use the same setting as in [3] for generating the box mask. The results are demonstrated in Figure. 7 and Table. 7 in Appendix Section A.1. Both quantitative results and qualitative results demonstrate that we are able to achieve better or comparable results compared to the baselines. We add an additional result section in the Appendix (Section A.1) to discuss about the results of the box-inpainting task. PSLD is able to generate very realistic images, but it sometimes has unstable reconstructions for box inpainting as we observed. We also attach the performance in Table II below.
>
> | Algorithm | PSNR | SSIM    | LPIPS |
> | -------- | ------- |------- |------- |
> | DPS |  22.85|0.861 | 0.127 |
> | DDRM | 24.33 | 0.860 | 0.120 |
> | DMPS | 22.01 |0.803 |0.233 |
> | PSLD | 23.99 | 0.787 |0.201 |
> | Latent-DPS | 23.14 | 0.784 |0.199 |
> | ReSample (Ours) | ***24.67*** | ***0.892*** |***0.093*** |
>
>
> Table II: Performance of box-inpainting on CelebA-HQ
>
> [1] Rombach, Robin, et al. "High-resolution image synthesis with latent diffusion models." Proceedings of the IEEE/CVF conference on computer vision and pattern recognition. 2022.
>
> [2] Rout, Litu, et al. "Solving linear inverse problems provably via posterior sampling with latent diffusion models." arXiv preprint arXiv:2307.00619 (2023).
>
> [3] Chung, Hyungjin, et al. "Diffusion posterior sampling for general noisy inverse problems." arXiv preprint arXiv:2209.14687 (2022).

---

> ### Author Response · Authors · 2023-11-22
> **Thank you for the review and any follow up questions?**
>
> Thanks again for your review! Your review is crucial for us to improve our manuscript.  Please let us know whether our response has addressed your concerns. If there is any remaining question about the manuscript and we will try our best to answer.

---

> > ### Comment · Reviewer_4YEX · 2023-11-22
> > **score updated**
> >
> > Dear authors,
> >
> > Thank you very much for addressing my questions and criticisms.
> >
> > The discussion about  the rationale behind why pixel-space diffusion models achieves lower quality than with our method is highly appreciated and valuable for future research.
> >
> > Also, I thank the authors for adding the new inpainting results with box type masks, which makes the paper stronger.
> >
> > Based on this, I decided to increase my score.
> >
> > Best

---

> ### Author Response · Authors · 2023-11-22
> **Thank you for the feedback!**
>
> Dear reviewer 4YEX,
>
> Thank you very much for the encouraging feedback! We are glad to see that you are satisfied with the response and improved the score of our paper, and willing to answer any further question.

---

### Official Review · Reviewer_azNZ · 2023-10-24

**Soundness:** 3 good
**Presentation:** 3 good
**Contribution:** 2 fair
**Rating:** 8
**Confidence:** 4

**Summary:**

In this paper, the authors propose a novel latent diffusion algorithm that ensures fidelity to the measurements for solving inverse problems. Instead of sampling in the image domain as often done in diffusion based methods, the sampling is performed in the latent domain which allows speeding up the algorithm. This approach is shown to perform well on several linear and non-linear inverse problems.

**Strengths:**

- The paper is overall well written and easy to follow.
- The background on denoising diffusion models is clear, giving the right balance between technicalities and intuitions.
- Experimental results are impressive. In particular, two points are to be underlined that make the method of interest for the imaging community:
  - The authors chose to work with (relatively) difficult measurement operators that may not be trivially SVD decomposable;
  - The authors considered mainly low measurement noise regime, which is a challenging setting for diffusion models but common setting in imaging.

**Weaknesses:**

- While the general background on diffusion models is well documented, key references are missing regarding diffusion models in constrained settings as well as latent space optimization methods (see references below).
- The comparison with some other methods seems rather unfair. In particular, ADMM-PnP is ran for only 10 iterations, while it often requires at least 50 to 100 iterations to yield good results; similarly, the FBP-UNet seems to strongly under-perform compared to the (Jin et al) reference.
- The motivations of the paper are not very consistent with the results (e.g. regarding the advantage of latent sampling on inference speed).

**Questions:**

**Major comments**

1. In the DPS paper, the authors do indeed compute $\nabla_z ||y-\mathcal{A}(\hat{x}_0(x_t)||_2^2$ in (7). However, their code does not use a squared norm, but a plain norm $\nabla_z ||y-\mathcal{A}(\hat{x}_0(x_t)||$ [(see here)](https://github.com/DPS2022/diffusion-posterior-sampling/blob/main/guided_diffusion/condition_methods.py#L32). In practice, changing to a squared norm strongly decreases the quality of the results. Is this also the case in this work? Would the theoretical analysis in the paper still hold?
2. It does not appear clearly in the paper that working in the latent domain improves the computational efficiency despite being a main motivation of this work by the authors. Table 7 is disappointing in that respect. Maybe the authors could emphasize other strengths of their method instead of this one (for instance, the successful results despite the difficulty to sample in constrained settings [1, 2, 3]).
3. An important issue in my opinion is the lack of reference to methods performing similar procedure as the proposed Hard Data Consistency. While diffusion models are overall well referenced, other methods are less well documented. In particular, optimizing latent variables of a decoder as in equation (10) is at the heart of well known methods such as [4, 5]. Discussing these references (and related ones) would help relate the author's work to methods not relying on diffusion. Other relevant works [1, 2, 3] could also be added to the references.
4. While the choice of solving (10) makes sense in practice, checking that the proposed sampling scheme works (or not) when (10) is solved at each iteration would be very insightful. So far, it is only backed by "Because of the continuity of the sampling process, after each data-consistency optimization step, the samples in the following steps can still remain similar semantic or structural information to some extent." How does your method perform if a data fidelity step was performed at each iteration?
5. The authors mention that (Rout et al., 2023) proposes a similar method; adding few sentences stressing the similarities and differences with the proposed method may be insightful.
6. The theoretical derivations in Appendix appear in a more general form in [6]. Adding the reference when introducing Tweedie's formula may be of interest.


**Minor comments**

6. Adding PSNR metrics when displaying images would be informative (e.g. in Fig. 3 and 4, but maybe more importantly in the appendix where there is no page limit).
7. Figure 6 is not very clear since the plot on the right does not correspond to the problem displayed on the left. Furthermore, the legend states "ReSample frequency" while the plot states "# of iterations", maybe the graph should either choose frequency in x-axis or the label should be updated.
8. Few typos are remaining, e.g. "in replace" (p. 5), "still remain" (instead of retain, p. 5), "noice" (p. 6).
9. If I understand Algorithm 1 correctly, the "if t \in C" condition is not activated before Stage 2 of Fig. 16. In this context, I do not understand how information about the measurement can be visible in Stage 1. Isn't Fig. 16 misleading with that regards?
10. More generally, is Stage 0 really necessary? I would naively that for solving eq. (10) and sampling from that may provide a good initial step, allowing to reduce the number of iterations.

**References**

[1] Guan-Horng Liu, Tianrong Chen, Evangelos A. Theodorou and Molei Tao, Mirror Diffusion Models for Constrained and Watermarked Generation arxiv:2310.01236, 2023

[2] Nic Fishman, Leo Klarner, Valentin De Bortoli, Emile Mathieu, and Michael Hutchinson, Diffusion models for constrained domains. arXiv preprint arXiv:2304.05364, 2023.

[3] Aaron Lou and Stefano Ermon. Reflected diffusion models. arXiv preprint arXiv:2304.04740, 2023

[4] Ulyanov, Dmitry, Andrea Vedaldi, and Victor Lempitsky. "Deep image prior." CVPR, 2018.

[5] Jalal, Ajil, Marius Arvinte, Giannis Daras, Eric Price, Alexandros G. Dimakis, and Jon Tamir. "Robust compressed sensing mri with deep generative priors." NeurIPS, 2021

[6] Efron, Bradley. "Tweedie’s formula and selection bias." Journal of the American Statistical Association 106.496 (2011): 1602-1614.

---

> ### Author Response · Authors · 2023-11-16
> **Official Response from authors of paper 4764 (Part I)**
>
> We would like to thank the reviewer for the insightful comments! Below, we address the reviewer’s concerns in detail:
>
> ***Comment:*** While the general background on diffusion models is well documented, key references are missing regarding diffusion models in constrained settings as well as latent space optimization methods
>
> ***Response:***  Thanks to the reviewer for sharing the reference list! We have added these references in our revised paper with more discussions about these methods in the introduction, methods and a new related work section in Appendix section E. We will add these references to the main paper if extra one page is provided during camera ready.
>
> ***Comment:*** The comparison with some other methods seems rather unfair. In particular, ADMM-PnP is ran for only 10 iterations, while it often requires at least 50 to 100 iterations to yield good results; similarly, the FBP-UNet seems to strongly under-perform compared to the (Jin et al.) reference.
>
> ***Response:*** Thanks for the question and we would like to clarify the baseline settings used in the paper as follows:
> 1. We only use 10 iterations because this achieves the best performance. We tune the ADMM-PnP parameters based on images in the validation set, and pick the hyperparameters that give the best performance. We observe that if we apply the pretrained denoiser with too many iterations  ( $>10$), it tends to over-smooth the image and cause a drop in performance.
>
> 2. For FBP-UNet, we are considering a much more challenging setting than [2]. In [2], they use 50 projections and no measurement noise. We experiment on 25 projections with additional Gaussian noise, which is much sparser. As a result, we can observe FBP gives severe streaking artifacts in this scenario as demonstrated in Figure .4 in our paper, which is not visible in [2]. This is the reason why FBP-UNet does not perform as good as the results reported in [2].
>
> To avoid the confusion, we have revised section C.3 in the Appendix to add more descriptions to clarify the baseline settings and discussions about the baseline results in the revised paper.
>
>
> ***Comment:*** Motivations of the paper not very consistent with results
>
> ***Response:*** We appreciate the reviewer’s feedback and the opportunity to clarify our paper’s motivation. We acknowledge that due to the hard data consistency update through the latent space, our algorithm requires slightly more inference time than several pixel-diffusion based approaches, while comes along with ***better memory efficiency*** than pixel-diffusion based methods as indicated in Table 4 in our paper.
>
> Because there are many challenges in using latent diffusion models for inverse problems, with one of the main challenges lying in the nonlinearity of the decoder, resulting in numerous local minima that lead to unfavorable results. This motivates us to propose the ReSample method to obtain reconstructions faithful to the measurements with latent diffusion models. To avoid confusion, we have strengthened our contribution of a novel methodology with satisfying performance in the introduction section (highlighted in color in the revised paper). We provide an example on the measurement consistency here in Table I, which demonstrates that our algorithm improves the measurement consistency a lot, which supports the our motivation. In this regard, we believe that our algorithm is an important first step in developing an efficient algorithm that can leverage latent diffusion models to solve inverse problems. We provide insights on this in Appendix section D.
>
> |Anatomical site|ReSample (Ours)|Latent-DPS|
> | -------- | ------- |------- |
> |Chest	|***5.36e-5***	|2.99e-3|
> |Abdominal|	***6.28e-5***|	3.92e-3|
> |Head	|***1.73e-5***	|2.03e-3|
>
> Table I: average measurement consistency loss for CT reconstruction for ReSample and Latent-DPS
>
>
>
> On the other hand, one of our primary motivations for this work is the computational demands associated with ***training*** diffusion models in pixel space as indicated in our paper's abstract and introduction, where latent diffusion models are able to achieve lower ***training*** time and reduced memory usage compared to pixel diffusion models, as observed and reported in prior works [3,4]. Experimental results are consistent with this motivation. We added another subsection in the computational efficiency section of the Appendix (Section A.5) to highlight this point about training efficiency. We also attach Table II below with a comparison of training time and memory of LDM [3] and DDPM [5] here.
>
>
> | Model type | Training Time | Training Memory | FID |
> | -------- | ------- |------- | ------- |
> | LDM | ***133 minutes***  |***7772MB*** |  ***100.48***   |
> | DDPM | 229 minutes | 10417MB | 119.50  |
>
>
> Table II: Efficiency and Performance of training 100k iterations of LDMs and DDPMs on CT images on a V100 GPU

---

> ### Author Response · Authors · 2023-11-16
> **Official Response from authors of paper 4764 (Part II)**
>
> ***Comment:*** Possible switch to a plain norm instead of square norm for optimization. Does the theoretical analysis still hold?
>
> ***Response:***  Thank you for this interesting observation! Following this comment, we conducted several studies on using a plain norm as our objective function instead of the squared norm. However, we did not observe much of a difference (if any) in the performance of ReSample. We conjecture that for DPS, the algorithm needs to compute the gradient with respect to the score function, whereas we calculate the gradient with respect to the decoder. Therefore, there may be some subtle differences in these two networks that favor a squared norm over a plain norm.
>
> In case of whether our theoretical analysis holds, our analysis mainly focuses on the resampling part of our algorithm, which takes an arbitrarily optimized $\hat{z}_0(y)$ as input, so the variance reduction and unbiasedness results still hold. Since the last theorem is studying unconditional reverse sampling and is not related to the hard consistency, we believe that it still holds for the plain norm.
>
>
>
> ***Comment:*** Emphasizing strengths other than computational efficiency
>
>
> ***Response:***  Thank you for the great suggestion! We acknowledge that due to the hard data consistency update through the latent space, our algorithm requires slightly more inference time than several pixel-diffusion based approaches, while comes along with ***better memory efficiency*** than pixel-diffusion based methods as indicated in Table 4 in our paper.  We edit our introduction part to emphasize more on our memory efficiency, the training efficiency of latent diffusion models and the novelty of our algorithm.
>
>
>
>
> ***Comment:*** Adding references and discussions about other methods such as CSGM
>
>
> ***Response:***  Thanks for catching this! We have added a related work section in our Appendix section E for a thorough discussion of other methods. We also include citations and a brief discussion of this in our introduction.
>
>
> ***Comment:*** How is our method performing when data fidelity is performed at every time step?
>
>
> ***Response:***  We also report additional results of both partial and full data-consistency performance in the task of chest CT reconstruction, as shown in Figure 18 and Table 8 in the Appendix section A.5 (shown as Table III here). Generally, from Table 8, we observe that when the skip step size (a skip step size of $10$ implies that we apply hard data consistency once every $10$ iterations of the reverse sampling process) get below 10, the performance almost saturates, which means we found that the quantitative metrics do not significantly change or only improve marginally with full data-consistency. Nevertheless, visual quality shows slightly sharper images with full data-consistency. We revise the ablation study section in the Appendix (Section A.5) to include both quantitative and qualitative results on this. This observation may imply that by ***only doing partial hard-consistency, we are able to achieve decent quality reconstructions*** that is not far from full data-consistency. This also shows the flexibility of our algorithms which can adjust the data consistency steps by balancing the trade-off between reconstruction quality and time cost.
>
> | k | PSNR | SSIM    |
> | -------- | ------- |------- |
> | 1 | 31.74| 0.922|
> | 4 | 31.74 | 0.923 |
> | 10 | 31.72 | 0.922 |
> | 20 | 31.23 | 0.918 |
> | 50 | 30.86 | 0.915 |
> | 100 | 30.09 | 0.906 |
> | 200 | 28.74 | 0.869 |
>
> Table III: average performance on chest CT images reconstruction with data consistency performed on each $k$ time step
>
> ***Comment:*** A discussion on the PSLD method [6]
>
> ***Response:***  Thanks for the suggestion! We have added a discussion a few sentences stressing the similarities and differences of our methods and [6] in the related work section in the Appendix (Section E) with a highlight of blue color.
>
>
> ***Comment:*** Adding references on discussing Tweedie's formula
>
> ***Response:***  Thanks for the suggestion! We added references on Tweedie's formula in our background section in the main text.
>
> ***Comment:*** Misinformation in Figure 16
>
> ***Response:***  Thanks for pointing this out! We remove this plot to remove the misinformation as in the chaotic stage we do not access the measurement (as in our experiments we only start to see patterns of the clean image after the middle of the second stage).

---

> ### Author Response · Authors · 2023-11-16
> **Official Response from authors of paper 4764 (Part III)**
>
> ***Comment:*** Is Stage 0 necessary
>
> ***Response:*** Thanks for pointing this out. We agree that existing works have achieved great progress on finding a good initial point/noise for reconstruction problems [7, 8]. However, it is still challenging to find the right time step to insert the noise, as some degradation is mild and some degradation is severe, as mentioned in [9]. Hence we resort to the method of starting from a random noise in our paper. Furthermore,  the approximation of $z_0$ via Tweedie’s formula may be inaccurate during the early sampling stages, potentially leading us to a poor local minimum. Thus, we believe that stage 0 is important for improving the performance of the current algorithm of ReSample. We agree with the reviewer that this is indeed a very interesting research question that could be a potential follow-up of our work in the future work.
>
>
> ***Comment:*** Other editing
>
> ***Response:***  We added the PSNR metrics on images for the additional results for box-inpainting and the ablation study on data consistency time steps in the Appendix A.2 and A.5. We modify Figure.6 to be more clear about the x-axis (i.e. ReSample frequency)
>
>
>
>
>
> [1] Ryu, Ernest, et al. "Plug-and-play methods provably converge with properly trained denoisers." International Conference on Machine Learning. PMLR, 2019.
>
> [2] Jin, Kyong Hwan, et al. "Deep convolutional neural network for inverse problems in imaging." IEEE transactions on image processing 26.9 (2017): 4509-4522.
>
> [3] Rombach, Robin, et al. "High-resolution image synthesis with latent diffusion models." Proceedings of the IEEE/CVF conference on computer vision and pattern recognition. 2022.
>
> [4] Luo, Simian, et al. "Latent Consistency Models: Synthesizing High-Resolution Images with Few-Step Inference." arXiv preprint arXiv:2310.04378 (2023).
>
> [5] Nichol, Alexander Quinn, and Prafulla Dhariwal. "Improved denoising diffusion probabilistic models." International Conference on Machine Learning. PMLR, 2021.
>
> [6] Rout, Litu, et al. "Solving linear inverse problems provably via posterior sampling with latent diffusion models." arXiv preprint arXiv:2307.00619 (2023).
>
>
> [7] Liu, Guan-Horng, et al. "I $^ 2$ SB: Image-to-Image Schr\" odinger Bridge." arXiv preprint arXiv:2302.05872 (2023).
>
>
> [8] Chung, Hyungjin, Jeongsol Kim, and Jong Chul Ye. "Direct Diffusion Bridge using Data Consistency for Inverse Problems." arXiv preprint arXiv:2305.19809 (2023).
>
> [9] Fabian, Zalan, Berk Tinaz, and Mahdi Soltanolkotabi. "Adapt and Diffuse: Sample-adaptive Reconstruction via Latent Diffusion Models." arXiv preprint arXiv:2309.06642 (2023).

---

> > ### Comment · Reviewer_azNZ · 2023-11-20
> >
> > I thank the authors for their clear and insightful answers and note the significant additional updates on their paper. I am satisfied with the response of the authors and am ready to improve my rating of the paper.
> >
> > One last comment (not requiring response from the authors) regarding ADMM-PnP: this algorithm is not state-of-the-art anymore, I encourage the authors to consider using the DPIR algorithm (https://github.com/cszn/DPIR) in their future works.

---

> ### Author Response · Authors · 2023-11-20
> **Thanks for the prompt response!**
>
> Dear reviewer azNZ,
>
> Thank you very much for the encouraging feedback! We are glad to see that you are satisfied with the response and willing to improve the rating of our paper.
>
>
> Thanks a lot for sharing the paper with us which looks very interesting! We will use the DPIR algorithm in our future work.

---

> ### Author Response · Authors · 2023-11-22
> **Thank you for your time and any additional questions?**
>
> Thanks again for your review and the response! Your review is crucial for us to improve our manuscript. If there is any remaining question about the manuscript and we will try our best to answer.

---

### Official Review · Reviewer_1zP2 · 2023-10-30

**Soundness:** 3 good
**Presentation:** 3 good
**Contribution:** 3 good
**Rating:** 6
**Confidence:** 3

**Summary:**

The paper titled: Solving Inverse Problems with Latent Diffusion Models Via Hard Data Consistency presents an novel algorithm that use pre-trained latent diffusion models to solve general inverse problems. The authors demonstrated the effectiveness on multiple tasks and datasets.

**Strengths:**

1. The authors propose a novel hard data consistency module, that strictly enforce the measurements via optimization, theoretical and empirical results demonstrate the superiority of ReSample over existing SOTA methods.

2. The authors prove theoretical proof with analysis on the variance of the resampled images.

3. The authors demonstrate the effectiveness of ReSample on multiple tasks on various benchmarks - deblurring, denoising, super resolution, inpainting CT Reconstruction.

4. This paper is well written and the core idea is clearly delivered.

**Weaknesses:**

I didn't find clear flaw of this paper, I have a few questions instead.

1. How do you scale up the algorithm to high-resolution images?

2. To solve equation 10 - the L2 least square, I assume you are using gradient descent or conjugate gradient descent, one possible way to accelerate it is to use close form expression on D(z) if A is a linear operator, Do you have any thoughts on this path?

3. Now, I assume the linear operator A is known, but for deblurring task, the system operator can be unknown, can you elaborate on how you could adopt your algorithms to this scenario?

**Questions:**

Please see above on my questions, overall a solid paper.

---

> ### Author Response · Authors · 2023-11-16
> **Official Response from authors of paper 4764**
>
> We would like to thank the reviewer for the insightful comments! Below, we address the reviewer’s concerns in detail:
>
> ***Comment:*** Scaling up to high-resolution images
>
> ***Response:*** To scale up the algorithm to high-resolution images, one possible way would be using an arbitrary-resolution (as long as divisible by 4 for instance) random noise as the initial noise of the LDM. Due to the convolutional nature of UNet of the LDM and the autoencoder, one can output arbitrary resolution generated images corresponding to the resolution of the latent vector, as demonstrated in [1]. In the revision, we have added this discussion in the Appendix section A.3 and provide an example of reconstructed high-resolution images in the task of random inpainting.
>
>
>
> ***Comment:*** Closed-form expression for data consistency
>
> ***Response:*** Thanks for pointing this out! We observe that in a noiseless setting with linear inverse problems, there is indeed a formula that would work, which is given by $\hat{z}_{0}(y) = E(D(\hat{z}_0) - (A^{+}AD(\hat{z}_0) - A^{+}y))$, where $E$ is the encoder. We revise the paper to provide an additional discussion on this formula in the Appendix section B. We also already applied a similar formula in the CT reconstruction task (where the pseudo-inverse is implemented by Conjugate Gradient) as demonstrated in Appendix B.1, and we revise this subsection for a clear description on how we use these formulas for our proposed optimization problem.
>
> ***Comment*** How to apply ReSample to an unknown linear operator
>
> ***Response:*** This is a very interesting question! To our best knowledge, we get to know there are some recent works [1,2,3,4] studying on the blind inverse problems with diffusion models. One interesting venue of these works is to train an image diffusion model and a parametrized operator diffusion model. At inference time, sampling is performed simultaneously from both diffusion models. Our algorithm can be incorporated with this kind of method by replacing the image diffusion model with the latent diffusion models. We leave this as an interesting follow-up research direction that we would like to explore more in the future.
>
> [1] Rombach, Robin, et al. "High-resolution image synthesis with latent diffusion models." Proceedings of the IEEE/CVF conference on computer vision and pattern recognition. 2022.
>
> [2] Chung, Hyungjin, et al. "Parallel diffusion models of operator and image for blind inverse problems." Proceedings of the IEEE/CVF Conference on Computer Vision and Pattern Recognition. 2023.
>
> [3] Murata, Naoki, et al. "Gibbsddrm: A partially collapsed gibbs sampler for solving blind inverse problems with denoising diffusion restoration." arXiv preprint arXiv:2301.12686 (2023).
>
> [4] Levac, Brett, Ajil Jalal, and Jonathan I. Tamir. "Accelerated motion correction for MRI using score-based generative models." 2023 IEEE 20th International Symposium on Biomedical Imaging (ISBI). IEEE, 2023.

---

> ### Author Response · Authors · 2023-11-22
> **Thank you for your time and any follow up questions?**
>
> Thanks again for your review! Your review is crucial for us to improve our manuscript.  Please let us know whether our response has addressed your concerns. If there is any remaining question about the manuscript and we will try our best to answer.

---

### Official Review · Reviewer_6mow · 2023-11-01

**Soundness:** 3 good
**Presentation:** 3 good
**Contribution:** 3 good
**Rating:** 8
**Confidence:** 4

**Summary:**

The paper proposes a latent-diffusion model based inverse problem solver which is more efficient than previous solves on pixel space and a novel problem.
The main contribution of the paper is an unbiased and non-expansive sampling method after applying data consistency, namely stochastic resample.
For various linear inverse problems, the proposed method has demonstrated comparable or better performance compared to baseline, while reducing memory and time cost.

**Strengths:**

- The paper is clearly written and easy to follow
- The proposed stochastic resampling is novel and theoretically solid.
- The performance of the proposed method outperforms baseline methods, especially simple extension of DPs.

**Weaknesses:**

- Complex usage of multiple engineering to improve the efficiency of the method (e.g. partial data consistency, early stopping...)

**Questions:**

- In the algorithm, at time point $t$, the data consistency update is done with $z_{t-1}$, rather than $z_t$ which is differ from prior works of diffusion-based inverse problem solver including DPS, DDNM and so on. In fact, to compute as in the algorithm, we need to feed-forward the diffusion model twice for each time step, which lead to two times longer sampling time. I think this is the reason that the paper emphasize the *partial* hard data-consistency update works well in the section 3.1. What if we use $z_t$ from the previous step without additional unconditional DDIM step? It seems reasonable because the mean of previous $z_t$ is also follows the unconditional sampling path. If I'm overlooking anything, please pointing it out.

- How long it takes if we apply the hard data-consistency for every time-steps? Also, can authors provide the quantitative result on it? I wonder how marginal the performance gap between full hard data-consistency and partial hard data-consistency (which is a proposed method) is, as the performance would be further improved if we use ReSample more frequently according to the figure 6 and figure 15.

- There is no description of $\gamma$ in figure 15

---

> ### Author Response · Authors · 2023-11-16
> **Official Response from authors of paper 4764**
>
> We would like to thank the reviewer for the insightful comments! Below, we address the reviewer’s concerns in detail:
>
> ***Comment:*** data consistency update is done with $z_{t-1}$ different from prior works
>
> ***Response:*** Sorry for the confusion. We have a slightly different notation from prior works. Prior works write the reverse sampling from $z_{t}$ to $z_{t-1}$ in their algorithm pseudo-code, whereas our algorithm denotes the reverse sampling from $z_{t+1}$ to $z_t$, and so indeed we only feed-forward the diffusion model once for each time step. Therefore, our work is actually doing data consistency updates on $z_t$, which is the same as in prior works. Figure 2 may illustrate this point more clearly with the blue arrows in the figure indicating the data consistency operations. Please let us know if this clarifies your comment, and we would be happy to follow up.
>
>
>
> ***Comment:*** Performance of full data-consistency on every time-step
>
> ***Response:*** We revised the ablation study section in the Appendix (Section A.5)  to report the inference time v.s. data consistency with different time steps of $k$ for chest CT reconstruction (we perform data consistency on every $k$ step, i.e. if $k=1$, we perform data consistency on every time step)
>
> We also report additional results of both partial and full data-consistency performance in the Table I below. We also revise the ablation study section in the Appendix (Section A.5) to include both quantitative and qualitative results on this, as shown in Figure 18 and Table 8. Generally, from Table I, we observe that when the skip step size (a skip step size of $10$ implies that we apply hard data consistency once every $10$ iterations of the reverse sampling process) get below 10, the performance almost saturates, which means we found that the quantitative metrics do not significantly change or only improve marginally with full data-consistency. Nevertheless, visual quality shows slightly sharper images with full data-consistency. We revise the ablation study section in the Appendix (Section A.5) to include both quantitative and qualitative results on this. This observation may imply that by ***only doing partial hard-consistency, we are able to achieve decent quality reconstructions*** that is not far from full data-consistency. This also shows the flexibility of our algorithms which can adjust the data consistency steps by balancing the trade-off between reconstruction quality and time cost.
>
>
> |Algorithm | k | PSNR | SSIM    | Inference Time|
> |---------| -----| ------- |------- |------- |
> | | 1 | 31.74| 0.922| 31:20 |
> | | 4 | 31.74 | 0.923 | 8:24 |
> |ReSample (Ours)| 10 | 31.72 | 0.922 | 3:52 |
> | | 20 | 31.23 | 0.918 | 2:16 |
> | | 50 | 30.86 | 0.915 | 1:20 |
> | | 100 | 30.09 | 0.906 | 1:08 |
> | | 200 | 28.74 | 0.869 | 0:54 |
>
>
> Table I: Inference times in mm:ss and performance for performing data consistency on each k step for chest CT reconstruction
>
>
>
> ***Comment:*** There is no description of $\gamma$ in figure 15
>
> ***Response:*** Thanks for pointing this out! We have added the description of $\gamma$ in Figure 15 in the revised manuscript.

---

> > ### Comment · Reviewer_6mow · 2023-11-20
> > **Thanks for the response**
> >
> > I appreciated the authors for their response and additional experiments.
> >
> > - Thank you for the clarification, but, I'm still confused. In algorithm 1, the proposed method seems to compute the score twice: $s_\theta(z_{t+1}, t+1)$ for unconditional sampling and $s_\theta (z'_t, t)$ for the hard data consistency. Is that just a typo?
> >
> > - Thanks for conducting additional experiments despite the short rebuttal period. It appears that ReSample with k=10 is sufficient for the CT reconstruction. Do the authors believe that this applies similarly to other images such as CelebA? No need to conduct more experiments; just consider whether the user can decide to use k=10 without additional parameter search or not.

---

> ### Author Response · Authors · 2023-11-20
> **Thanks for the prompt response!**
>
> Dear reviewer 6mow,
>
> Thanks for the prompt response!
>
> >the proposed method seems to compute the score twice: for unconditional sampling $s_\theta (z_{t+1}, t+1)$ and for the hard data consistency $s_\theta (z'_t, t)$. Is that just a typo?
>
> Thank you for catching this! We use the $s_\theta (z_{t+1}, t+1)$ for both unconditional sampling and hard data consistency. We just uploaded a revised version of the paper to make it clear that we only compute the score once per each time step (we also highlight our resample part of the algorithm in blue) at the top of page 5.
>
> >Do the authors believe that this applies similarly to other images such as CelebA?
>
> This is a great question! Indeed we use $k=10$ for ***every*** experiment we report in this work including CelebA-HQ, FFHQ, and CT images. We observe that $k=10$ works well for every experiment we reported in this work while striking a good balance between inference efficiency and reconstruction quality. Intuitively, the next couple steps of samples after the data consistency step can retain similar semantic information to some extent, so we guess that a mild partial consistency like $k=10$ should be applicable to a variety of situations.

---

> ### Author Response · Authors · 2023-11-22
> **Thank you for your time and any follow up questions?**
>
> Thanks again for your review and your follow-up discussion! Your review is crucial for us to improve our manuscript. Feel free to let us know if there is any remaining question about the manuscript and we will try our best to answer.

---

### Author Response · Authors · 2023-11-16
**Global Response from authors**

Firstly, we would like to sincerely thank the reviewers for taking the time to review our paper and for providing constructive feedback. We are encouraged that the reviewers think highly of our work in the sense that "the proposed stochastic resampling is novel and theoretically solid" (Reviewer 6mow), "this paper is well written and the core idea is clearly delivered" (Reviewer 1zP2), "experimental results are impressive" (Reviewer azNZ), and that we provide "a design of a novel sampling method supported by theory." (Reviewer 4YEX).
First, we address the common questions raised by all reviewers, highlighting several key points in our work and paper revision.


***Comment 1:*** Motivations of the paper not very consistent with the results.

***Response:*** We appreciate the valuable feedback from the reviewers and the opportunity to clarify the motivation behind our paper. It is important to note that, due to the hard data consistency update through the latent space, our algorithm requires slightly more inference time than several pixel-diffusion-based approaches. However, our method demonstrates superior memory efficiency compared to pixel-diffusion-based methods, as indicated in Table 4 in our paper.
There are many challenges in using latent diffusion models for inverse problems, with one of the main challenges lying in the nonlinearity of the decoder, resulting in numerous local minima that lead to unfavorable results (we provide insights on this in Appendix section D). This motivates us to propose the ReSample method to obtain reconstructions faithful to the measurements with latent diffusion models. We demonstrate that our method outperforms many of the provided baselines through qualitative and quantitative results. We believe that our paper represents a significant advancement in efficiently solving inverse problems with latent diffusion models.

Lastly, to clarify the motivation behind our work, we emphasize our goal of addressing the computational demands associated with ***training*** diffusion models in pixel space. This motivation is clearly outlined in the abstract and introduction, where latent diffusion models are shown to achieve lower ***training*** time and reduced memory usage compared to pixel diffusion models, as reported in prior works [1,2]. Experimental results consistently support this motivation, where we have also added a subsection in the Appendix (i.e., Appendix A.5) to underscore this point about training efficiency.


***Comment 2:*** More challenging inpainting experiments (box inpainting)

***Response:*** We conduct additional experiments for the box inpainting task on the CelebA-HQ dataset, employing the same settings as in [3] to generate the box mask. Results are presented in Figure 7 and Table 7, showcasing both quantitative and qualitative improvements compared to the baselines. Further details on the results of the box-inpainting task can be found in an additional section in the Appendix (Section A.5).


***Comment 3:*** Performance on performing hard data consistency on every time step

***Response:*** We also report additional results on both partial and full hard data consistency performance in the task of chest CT reconstruction, as depicted in Figure 18 and Table 8. Generally, from Table 8, we observe that when the skip step size (a skip step size of $10$ implies that we apply hard data consistency once every $10$ iterations of the reverse sampling process) get below 10, the performance almost saturates, indicating that quantitative metrics do not significantly change or only show marginal improvement with full data-consistency. In the meanwhile, the visual quality exhibits slightly sharper images with full data-consistency.
Based upon the reviewer’s suggestion, we have revised the ablation study section in the Appendix (Appendix A.5) to include both quantitative and qualitative results on this. This observation implies that by only implementing partial hard-consistency, we can achieve decent quality reconstructions that are not far from full data-consistency. This also demonstrates the flexibility of our algorithms, which can adjust data consistency steps, striking a balance between reconstruction quality and time cost.


[1] Rombach, Robin, et al. "High-resolution image synthesis with latent diffusion models." Proceedings of the IEEE/CVF conference on computer vision and pattern recognition. 2022.

[2] Luo, Simian, et al. "Latent consistency models: Synthesizing high-resolution images with few-step inference." arXiv preprint arXiv:2310.04378 (2023).

[3] Chung, Hyungjin, et al. "Diffusion Posterior Sampling for General Noisy Inverse Problems." The Eleventh International Conference on Learning Representations. 2022.

---

### Meta-Review · Area_Chair_J8QE · 2023-12-10

**Metareview:**

The work proposes to solve inverse problems with a diffusion process done in the latent space (before the decoder network).
The paper is evaluated on multiple imaging inverse problems and comes with a theoretical analysis of the variance reduction and the bias of the model.

The paper was considered well written and as a novel contribution on the SoTA.

All reviewers and the AC endorse the paper for publication.

**Justification For Why Not Higher Score:**

The paper is considered novel but not a significant improvement with the potential to challenge the SoTA.

**Justification For Why Not Lower Score:**

All reviewers and the AC endorse the paper for publication.

---

### Decision · Program_Chairs · 2024-01-16

Accept (spotlight)